# A linear response framework for quantum simulation of bosonic and fermionic correlation functions

Efekan Kökcü ●[1] ✉, Heba A. Labib ●[1], J. K. Freericks ●[2] & A. F. Kemper ●[1] ✉

Response functions are a fundamental aspect of physics; they represent the link between experimental observations and the underlying quantum many-body state. However, this link is often under-appreciated, as the Lehmann formalism for obtaining response functions in linear response has no direct link to experiment. Within the context of quantum computing, and via a linear response framework, we restore this link by making the experiment an inextricable part of the quantum simulation. This method can be frequency- and momentum-selective, avoids limitations on operators that can be directly measured, and can be more efficient than competing methods. As prototypical examples of response functions, we demonstrate that both bosonic and fermionic Green's functions can be obtained, and apply these ideas to the study of a charge-density-wave material on the *ibm_auckland* superconducting quantum computer. The linear response method provides a robust framework for using quantum computers to study systems in physics and chemistry.

Quantum computers are showing promise as quantum simulators of many-body physics, with the hope of being able to further our understanding of complex interacting systems. In order to realize this promise, a key task is to compute response functions for a prepared many-body state. They represent the experimental measurements that are performed on the physical realizations of such systems, and computing them via simulation is a critical step in connecting to experiments and building an understanding of the physics they contain. Examples of experiments that measure response functions are neutron scattering, optical spectroscopy, and angle-resolved photo-emission spectroscopy (ARPES), which measure the spin-spin correlation, current-current correlation, and single-particle Green's function, respectively[1,2]. The first two are bosonic correlation functions, while the latter is a fermionic correlation function. Both of these contain valuable information – both have direct links to experiments, and in addition the electronic Green's function is a key ingredient in hybrid-classical algorithms such as dynamical mean field theory[3–8].

In all cases, correlation functions involve expectation values of the form $\langle \mathbf{A}(t)\mathbf{B}(t')\rangle$; the particulars of **A** and **B** are set by the experiment or desired quantity. For example, in many condensed matter scattering experiments both **A** and **B** have definite momentum, i.e. a sum of local operators; in contrast, scanning probe microscopy makes use of purely local operators. This means that there is significant variation in **A** and **B**, and in turn the need for a comparable freedom in evaluating these on the quantum computer.

There are several existing techniques for computing correlation functions on quantum computers. Primary among these are methods based on the Hadamard test circuit, which rely on time evolution of a given state[9–16]; other methods (most of which rely on the Lehmann formalism[1,17]) include variational approaches[18–22], spectral decomposition[23–25], and linear systems of equation solvers[26]. Each of these has their own advantages and disadvantages, based on the particular quantum algorithms and hardware at hand. Moreover, extending beyond simple local unitary **A** and **B** comes with the cost of additional resources and increased error in the calculations.

In this work, we outline a method for calculating correlation functions based on a linear response framework that is in direct correspondence to experiments, as schematically illustrated in Fig. 1. The quantum state is driven with an applied field **B** with specific temporal and spatial structure, and the response of the system to that field **A** is

[1]Department of Physics, North Carolina State University, Raleigh, NC 27695, USA. [2]Department of Physics, Georgetown University, 37th and O Sts. NW, Washington DC, WA 20057, USA. ✉e-mail: ekokcu@ncsu.edu; akemper@ncsu.edu

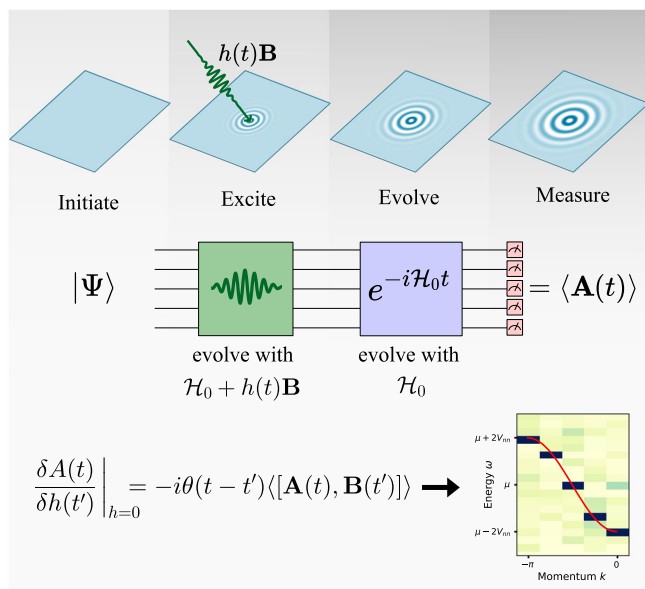

**Fig. 1 | Linear response method.** We establish an equivalence between the experimental measurement of a response function and an ancilla-free quantum simulation under a time dependent Hamiltonian that includes the perturbative excitation $h(t)\mathbf{B}$. Following excitation, the system is evolved under $\mathcal{H}_0$, and $\mathbf{A}$ is measured. The functional derivative of $A(t) = \langle \mathbf{A}(t) \rangle$ with respect to $h(t')$ yields the retarded response function shown in the figure. The data shown is taken from Fig. 3.

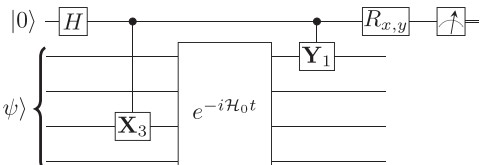

**Fig. 2 | Hadamard test circuit structure.** for obtaining the real and imaginary parts of the correlation function $\langle \psi | Y_1(t) X_3 | \psi \rangle$. If the final rotation is $R_x$ ($R_y$), the real (imaginary) part is measured.

been explored in this realm. Our work here does not focus on this classical application, but it should be clear that the approach developed here can be directly applied more broadly.

## Results

To set the stage for our discussion, we will briefly outline the details of the Hadamard test[9] approach for obtaining correlation functions, which underpins several of the currently used methods[10–16] on near term quantum hardware, and shares the most characteristics with the linear response framework. The other methods are either variational[18–22], or are not amenable to near-term quantum hardware[23–25].

Figure 2 illustrates the circuit structure used for the Hadamard test. To measure $\langle \psi | \mathbf{A}(t) \mathbf{B} | \psi \rangle$ with Hadamard test method, one needs to apply $\mathbf{A}$ and $\mathbf{B}$ in a controlled fashion, i.e. apply them only if the ancilla qubit is in $|1\rangle$ state. It is efficient to do so when $\mathbf{A}$ and $\mathbf{B}$ are unitary operators that are local in qubit space, but when they are not, further resources are needed.

As an example, consider a correlation function in which the operator $\mathbf{A}$ is a linear combination of $n$ Pauli strings where $n$ is the number of qubits, $\mathbf{A} = \sum_{i=1}^{n} \zeta_i X_i$, and $\mathbf{B}$ is still local and unitary. One can calculate this correlation function within this framework in two ways. The first way is to run several circuits to obtain correlation functions $\langle \psi | X_i(t) \mathbf{B} | \psi \rangle$, and post-process the results. Further, we show that this approach is not quantum hardware noise-robust, and generates significantly more error compared to our (second) method. In addition, the number of shots required for this method scales with the number of local operators in the operators. It is simply because to ensure an error threshold $\epsilon$ for a linear combination of $n$ terms, one must set the error for each term to $\epsilon/n$. This in the end leads to a number of shots $N_{shot} = \mathcal{O}(n^2/\epsilon^2)$ per term, scaling quadratically with the system size. The second way is to implement the Linear Combination of Unitaries (LCU)[32], to apply $\mathbf{A}$ directly on the state. For the case where $\mathbf{A}$ is not unitary, the LCU becomes a probabilistic algorithm with a success rate that decreases exponentially with the system size $n$. This can be cured for the case where the operator is unitary by Oblivious Amplitude Amplification, but at the cost of significantly increased circuit depth[33].

The correlation functions we consider are of the form $\langle \mathbf{A}(t) \mathbf{B}(t') \pm \mathbf{B}(t') \mathbf{A}(t) \rangle$ — the amplitude of the operator $\mathbf{A}$ at time $t$ given that $\mathbf{B}$ acted on the system at time $t'$. The operators can be chosen to be local in position or momentum space to obtain spatial information about the system. The amplitudes are substracted in the case of bosonic correlation functions, whereas they are added in the case of fermionic correlation functions. Both can be calculated via the linear response method that we present here. We will first describe the formalism for bosonic correlation functions and describe how to apply momentum and frequency selectivity. Then, we describe two different ways to apply the linear-response formalism to calculate fermionic correlation functions for Hamiltonians that conserve particle count parity (maintain even or odd numbers of electrons). In what follows, we assume the existence of a pure or mixed state of interest on the quantum computer, which was prepared for example via variational approaches or adiabatic state preparation. In addition, we assume access to a method to implement evolution under both time-

measured as a function of space and time. The proportionality between the field and the response then yield the desired correlation function(s).

The linear response framework enables using generic operators $\mathbf{A}$ and $\mathbf{B}$ (such as linear combinations) in the correlation function within a single quantum circuit, which has several downstream advantages. (i) Many correlation functions may be obtained at the same time, with one quantum circuit. (ii) Relying on a single quantum circuit avoids compounding errors in post-processing. (iii) Tailored excitations, e.g. those that focus on a particular energy or momentum range of the correlation function can be made. (iv) No ancilla qubits are needed. While more complex operators are also possible within the competing frameworks such as the Hadamard test, as it will be mentioned further, it comes at a significantly large additional cost.

We demonstrate the power of the linear response framework by applying it to the study of a two model systems. The first is a prototypical charge density wave system — the Su–Schrieffer–Heeger model[27]. We use the two fermionic methods with a momentum-selective field to obtain the electronic spectrum as would be measured by ARPES on IBM quantum hardware, and on a noisy simulator to compare the linear-response method to the Hadamard-test method. We next use the bosonic method and frequency selectivity to obtain the density-density response function of the same model system, as would be measured by momentum-resolved electron energy loss spectroscopy (M-EELS). Finally, we demonstrate the use of the technique for interacting models and compute the spectrum (retarded Green's function) of the 1D Hubbard model. These developments make significant inroads to being able to use near-term quantum computers in real-world applications.

This work also has impact on classical computing; the approach described below allows for one to compute response functions by simply running time evolution on a classical computer which avoids the need to compute vertex corrections by solving a Bethe–Salpeter-like equation, as needed with conventional approaches[2]. While our work has existing analogs for bosonic correlation functions[28–31], to the best of our knowledge the fermionic correlation functions have not yet

independent and time-dependent Hamiltonians. There are many choices for both; the complexities of these methods and others are detailed elsewhere[34].

## Bosonic (commutator) correlation functions

The methodology employs the standard results from linear response in many-body physics (see e.g. refs. 2,17 and [1]), as we develop below. We are interested in the expectation value of the operator $\mathbf{A}(t)$ measured in a prepared many-body state $|\psi_0\rangle$ and time-evolved in the Hamiltonian plus the applied (Hermitian) field; $h(t)\mathbf{B}$ that is, $\mathcal{H}(t) = \mathcal{H}_0 + h(t)\mathbf{B}$. Then, $A(t)$ is given by

$$A(t) = \langle\psi_0|U(t)^\dagger \mathbf{A} U(t)|\psi_0\rangle \qquad (1)$$

$$U(t) = \mathcal{T}_t e^{-i\int_{t_s}^{t}[\mathcal{H}_0 + \mathbf{B}h(\bar{t})]d\bar{t}}, \qquad (2)$$

where $U(t)$, in Eq. (2), is the time ordered exponential for time evolution with respect to the time-dependent Hamiltonian plus field, and $t_s$ is the starting time such that $t > t_s$, and $h(t') = 0$ for all $t' < t_s$. Expanding $A(t)$ with respect to $h(t)$, we find

$$A(t) = \int dt' \chi^R(t,t')h(t') + \mathcal{O}(h^2). \qquad (3)$$

Here, $\chi^R(t,t')$ is defined to be the functional derivative of $A(t)$ with respect to $h(t')$. Using Eqs. (1) and (2), we obtain

$$\left.\frac{\delta A(t)}{\delta h(t')}\right|_{h=0} = -i\theta(t-t')\langle\psi_0|[\mathbf{A}(t),\mathbf{B}(t')]|\psi_0\rangle, \qquad (4)$$

where we have $\mathbf{A}(t) := e^{it\mathcal{H}_0}\mathbf{A}e^{-it\mathcal{H}_0}$. The $\theta$-function arises because in Eq. (2) the integration region on the time ordered exponents is limited to $\bar{t}$ values that are smaller than $t$. Since $\mathcal{H}_0$ is time independent, the response function $\chi^R(t,t')$ only depends on the time difference $t - t'$. Fourier transforming from time to frequency, and using the convolution theorem, yields

$$A(\omega) = \chi^R(\omega)h(\omega) + \mathcal{O}(h^2). \qquad (5)$$

Thus, if the amplitude of the signal $h(t)$ is chosen to be small enough, the higher-order terms can be neglected and the response function can be calculated as a simple ratio in the frequency basis. Precisely how small the amplitude should be can be found in SI, and will be mentioned in the following sections.

One might be interested in the response function centered in a specific frequency interval and want to improve the signal-to-noise ratio of the calculation. This is achieved by choosing the frequency support of $h(t)$ to be most concentrated within the desired frequency interval.

Similarly, by choosing $\mathbf{A}$ and $\mathbf{B}$ as operators with definite momentum, we can directly calculate the response function in the momentum basis. For example, for creation of a magnon with momentum $k$ we can pick $\mathbf{B} = \sum_r e^{ikr}X_r + \text{H.C.} = \sum_r 2\cos(kr)X_r$, where $X_r$ is Pauli $X$ matrix applied on the $r$th site. In general, $\mathbf{B} = \sum_r \zeta_r \sigma_r$ will be a linear combination of non-commuting Pauli strings. In that case, the signal can be implemented via 1st order Trotter-Suzuki approximation

$$e^{-ih(t)\mathbf{B}\Delta t} = \prod_r e^{-ih(t)\Delta t \zeta_r \sigma_r} + \mathcal{O}(h^2). \qquad (6)$$

For further error analysis, see SI. We can use a similar form for $\mathbf{A}$ as we use for $\mathbf{B}$, but since it is directly measured (rather than appearing in the time evolution), this can simply be achieved with multiple circuits. However, if $\mathcal{H}_0$ is translation invariant, a single circuit is sufficient to calculate the response function $\chi^R$ in momentum space for a given momentum value, because it satisfies

$$\chi^R_{k,k'}(t-t') = \delta_{k,k'}\chi^R_{k,k}(t-t'); \qquad (7)$$

that is, it is diagonal in momentum. This property is purely a result of translational invariance: momentum is always conserved. and thus unless $k = k'$, the amplitudes must be zero, leading to Eq. (7).

Momentum and frequency selectivity allow us to immediately focus the signal we obtain from the quantum computer into desired ranges of momentum or frequency. This frequency selectivity is not easily performed with the Hadamard test[35,36]. Moreover, implementing a momentum selective operator can only be achieved via costly circuit modifications such as LCU. To avoid this, other approaches require each real space pair $(r_1, r_2)$ to be measured separately with independent circuits; these are then Fourier transformed to obtain a momentum response function. On a noisy device, errors from the different circuits measurements can lead to both structured and unstructured noise (see Supplementary Note 2), reducing the precision of the final result. In the following sections, we show that momentum selectivity in our approach significantly reduces noise in the measured signal.

In short, the procedure to obtain the correlation function $\langle\psi_0|[\mathbf{A}(t), \mathbf{B}]|\psi_0\rangle$ is as follows:

1. Evolve $|\psi_0\rangle$ with the perturbed Hamiltonian $\mathcal{H}(t) = \mathcal{H}_0 + h(t)\mathbf{B}$ during the time where $h(t)$ is finite. $h(t)$ should be a small field in order to ensure the simulation is in the linear response regime.
2. Continue to evolve with the unperturbed Hamiltonian $\mathcal{H}$. The maximum length of time needed is set by the desired minimum energy resolution.
3. At each time of interest $t$, measure $A(t) = \langle\mathbf{A}(t)\rangle$.
4. Fourier transform $A(t)$ to $A(\omega)$ and divide by $h(\omega)$ to obtain $\chi(\omega)$, thus performing the (numerical) functional differentiation.

## Fermionic (anti-commutator) correlation functions

The most important fermionic correlation function is the retarded electronic Green's function given by

$$G^R(r_i,t;r_j,t') = -i\theta(t-t')\langle\psi_0|\{c_i(t), c_j^\dagger(t')\}|\psi_0\rangle, \qquad (8)$$

where $c_i$ and $c_j^\dagger$ are the fermionic annihilation and creation operators at $r = r_i$ and $r_j$, respectively. Note that Eq. (8) is the correlation function with respect to a single many-body state $|\psi_0\rangle$. For the Green's function at $T = 0$ in standard many-body theory $|\psi_0\rangle$ is the ground state. At finite temperatures the expectation value has to be additionally averaged over a thermal distribution of states, which can be achieved via classical averaging of eigenstates[37–39] or by going over to a density matrix representation[32,40–45]. The formalism below is applicable for any of these cases.

The functional derivative method does not directly carry over, because it requires adding a Grassman number valued field, which cannot be easily realized in a numerical simulation. This has thus far limited the potential of ancilla-free methods to bosonic correlation functions only[35,36]. To overcome this, we introduce two complementary approaches. The first uses an auxiliary operator $\mathbf{P}$ which anti-commutes with $\mathbf{B}$, while the second uses simple post-selection.

First, we discuss a method based on the use of an auxiliary operator. We consider the fermionic version of Eq. (4), and denote this by $G(t,t')$:

$$G(t,t') = -i\theta(t-t')\langle\psi_0|\{\mathbf{A}(t), \mathbf{B}(t')\}|\psi_0\rangle. \qquad (9)$$

In order to produce an anticommutator, we introduce an additional operator $\mathbf{P}$ which satisfies the following properties

1. $\mathbf{P}|\psi_0\rangle = s|\psi_0\rangle$ with $s \neq 0$.
2. $\{\mathbf{B}(t), \mathbf{P}\} = 0$ for all times $t$.

3. $[\mathcal{H}_0, \mathbf{P}] = 0$, or $\mathbf{P}$ has no time dependence.

With these properties, it is straightforward to show that

$$G(t,t') = \frac{i}{s}\theta(t-t')\langle\psi_0|[\mathbf{A}(t)\mathbf{P}(t), \mathbf{B}(t')]|\psi_0\rangle. \qquad (10)$$

This is of the form of Eq. (4) with $\mathbf{A}(t)$ replaced by $\mathbf{A}(t)\mathbf{P}(t)$; therefore, the bosonic linear response method can be directly used.

When $G(t,t')$ is the retarded electronic Green's function Eq. (8), the assumptions are satisfied by the parity operator for Hamiltonians that preserve particle parity; this covers a vast class of Hamiltonians of interest in quantum chemistry, condensed matter physics and quantum field theory. If the Hamiltonian of interest conserves the parity of the electron number, then the parity operator $\mathbf{P} = Z_1 Z_2 \ldots Z_n$ satisfies second and third conditions, where we use the spin representation (obtained after Jordan-Wigner transformation) to represent the parity operator. The fermionic operators, $c_i$ and $c_i^\dagger$, in their spin representation, have a Jordan-Wigner string attached[46]; that is, they are composed of $i-1$ consecutive $Z$ operators followed by a $X \pm iY$:

$$c_i = Z_0 Z_1 \ldots Z_{i-1}\frac{X_i - iY_i}{2} = \frac{\tilde{X}_i - i\tilde{Y}_i}{2} \qquad (11)$$

$$c_i^\dagger = Z_0 Z_1 \ldots Z_{i-1}\frac{X_i + iY_i}{2} = \frac{\tilde{X}_i + i\tilde{Y}_i}{2}. \qquad (12)$$

In this case both $c_i$ and $c_i^\dagger$ anticommute with the parity operator $\mathbf{P} = Z_1 Z_2 \ldots Z_n$, which satisfies the second condition. With this, $G(t,t')$ can be obtained by measuring Eq. (10) upon replacing $\mathbf{A}$ with $\tilde{X}_i\mathbf{P}$ (and/or $\tilde{Y}_i\mathbf{P}$) and $\mathbf{B}$ with $\tilde{X}_j$ (and/or $\tilde{Y}_j$).

While the requirements on $\mathbf{P}$ may seem restrictive, particle number parity conserving Hamiltonians form a large class containing many systems of interest. First, it covers any particle number conserving system such as molecules and condensed matter systems such as Hubbard model; all terms of these Hamiltonians have equal number of creation and annihilation operators ($c_i^\dagger c_j$, $c_i^\dagger c_j^\dagger c_r c_s$, ...). In addition to these systems, this method of auxiliary operator works for Hamiltonians that contain pair creation/annihilation terms such as $c_i c_j$, $c_i^\dagger c_j^\dagger$. These clearly do not conserve particle number; however, these terms generate or destroy even number of particles, leading to the conservation of the particle number parity operator. Similar terms are present in the effective theories for superconductivity.

We can choose $h(t)$ and $\mathbf{B}$ to have frequency and momentum selectivity in the same way as we did for bosonic correlation functions. Thus, we can directly calculate the fermionic Green's function in momentum space,

$$G^R(k,t;k',t') = -i\theta(t-t')\langle\psi_0|\{c_k(t), c_{k'}^\dagger(t')\}|\psi_0\rangle, \qquad (13)$$

by selecting $\mathbf{A}$ as a Fourier combination of $\tilde{X}_i\mathbf{P}$ (and/or $\tilde{Y}_i\mathbf{P}$) with momentum $k$, and $\mathbf{B}$ as a Fourier combination of $X_j$ (and/or $Y_j$) with momentum $k'$, and forming the appropriate linear combination to select the desired $c/c^\dagger$ terms. Similarly, by choosing an appropriate frequency support for $h(t)$, we can calculate $G^R$ in a desired frequency range.

We next turn to a post-selection method to obtain fermionic single-particle Green's functions. When the desired anti-commutator is the single-particle Green's function (Eq. (8)) for a particle number conserving Hamiltonian, i.e. $|\psi_0\rangle$ is an $N$-particle wave function, a powerful alternate approach exists. A complete derivation is shown in the supplementary material; we outline the salient parts here. Let us

specify our perturbing field as

$$\mathbf{B} = \sum_m \alpha_m \tilde{X}_m = \sum_m \alpha_m(c_m + c_m^\dagger). \qquad (14)$$

Position or momentum selectivity can be imposed by the choice of $\alpha_m$. Starting from a wavefunction with $N$ particles and evolving with $\mathcal{H}_0 + h(t)\mathbf{B}$, the system will be in a superposition of the $N-1$, $N$, and $N+1$ particle sectors to linear order in $h(t)$. For clarity, let us choose $h(t) = \eta\delta(t)$ where $\eta \ll 1$ and $\delta(t)$ is a Dirac delta pulse. This choice is not necessary, we can choose $h(t)$ more generally to achieve frequency selectivity. In order to measure the Green's function we apply a rotation about $y$ (or $x$) to enable measurement of $c_1 \pm c_1^\dagger$ on the first qubit, which generates $N-2$ and $N+2$ particle states as well. Denoting $|\Phi_M^y\rangle$ (or $|\Phi_M^x\rangle$) as the $M$ particle component of this final state, the state right before the measurement with $y$ rotation is

$$|\Psi^y\rangle = |\Phi_{N-2}^y\rangle + |\Phi_{N-1}^y\rangle + |\Phi_N^y\rangle + |\Phi_{N+1}^y\rangle + |\Phi_{N+2}^y\rangle,$$

and with $y \leftrightarrow x$ for the $x$ rotation case. The $|\Phi_M^{x(y)}\rangle$ components of the final state are not normalized, and in fact their norms give the probability to observe $M$ particles. These components can be separated via post selection, and quantities such as $\langle\Phi_M^y|\Phi_M^y\rangle$ and $\langle\Phi_M^y|c_1^\dagger c_1|\Phi_M^y\rangle$ can be measured within the $M$ particle sectors. In Supplementary Note 3, we show that the following linear combinations of those quantities yield the desired Green's functions:

$$\langle\Phi_{N-1}^y|\Phi_{N-1}^y\rangle + \langle\Phi_{N+1}^y|\Phi_{N+1}^y\rangle = \frac{1}{2} + \eta\sum_m \alpha_m \text{Re}\left[G_{1m}^>(t) - G_{1m}^<(t)\right]$$
$$= \frac{1}{2} + \eta\sum_m \alpha_m \text{Re}\, G_{1m}^R(t) \qquad (15)$$

$$\langle\Phi_N^y|c_1^\dagger c_1|\Phi_N^y\rangle + \langle\Phi_{N+1}^y|\Phi_{N+1}^y\rangle = \frac{1}{2} + \eta\sum_m \alpha_m \text{Re}\left[G_{1m}^>(t) + G_{1m}^<(t)\right] \qquad (16)$$

$$\langle\Phi_{N-1}^x|\Phi_{N-1}^x\rangle + \langle\Phi_{N+1}^x|\Phi_{N+1}^x\rangle = \frac{1}{2} + \eta\sum_m \alpha_m \text{Im}\left[G_{1m}^>(t) - G_{1m}^<(t)\right]$$
$$= \frac{1}{2} + \eta\sum_m \alpha_m \text{Im}\, G_{1m}^R(t) \qquad (17)$$

$$\langle\Phi_N^x|c_1^\dagger c_1|\Phi_N^x\rangle + \langle\Phi_{N+1}^x|\Phi_{N+1}^x\rangle = \frac{1}{2} + \eta\sum_m \alpha_m \text{Im}\left[G_{1m}^>(t) + G_{1m}^<(t)\right] \qquad (18)$$

where the fermionic Green's functions are[1],

$$G_{ij}^<(t) = i\langle\psi_0|c_j^\dagger(0)c_i(t)|\psi_0\rangle$$
$$G_{ij}^>(t) = -i\langle\psi_0|c_i(t)c_j^\dagger(0)|\psi_0\rangle \qquad (19)$$
$$G_{ij}^R(t) = -i\theta(t)\langle\psi_0|\{c_i(t), c_j^\dagger(0)\}|\psi_0\rangle.$$

While this is limited to particle-conserving Hamiltonians, this is a relatively mild restriction as all fermionic Hamiltonians that do not have superconducting terms (pair-creation and pair-annihilation) satisfy this restriction.

## Error analysis and scalability

In Supplementary Note 6, we analyze three different sources of error for a time local signal $h(t) = \eta\delta(t)$ in detail: non-linear excitations, Trotter error for the driving field $e^{i\eta\mathbf{B}}$, and the statistical error coming from measurement. To ensure that the non-linear contribution is smaller than $\epsilon_{NL}$, the signal amplitude should be chosen as $\eta = \mathcal{O}(\epsilon_{NL}/||\mathbf{B}||^2||\mathbf{A}||)$, where $||.||$ is the spectral norm. We further show that $1^{st}$ order Trotter error is $\mathcal{O}(\epsilon_{NL}\alpha_{comm}^1(\mathbf{B})/||\mathbf{B}||r)$ where $r$ is the number of Trotter steps

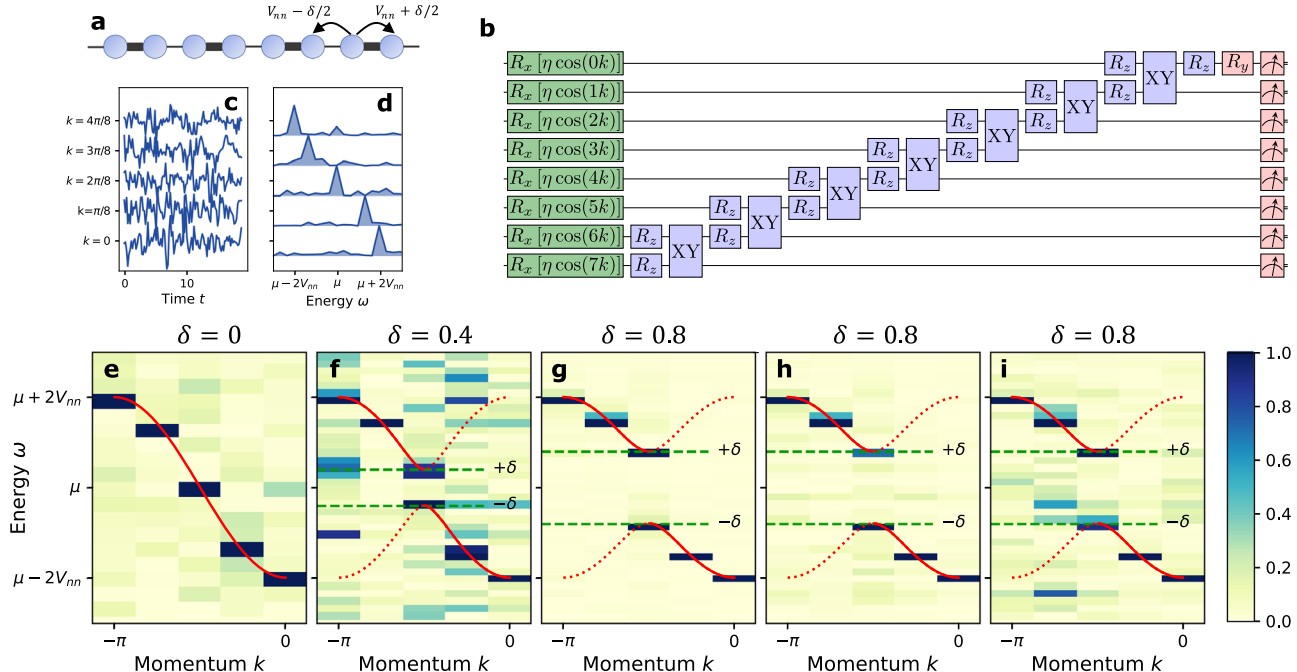

**Fig. 3 | Electronic Green's function for the Su–Schrieffer–Heeger (SSH) model. a** Lattice and hopping structure of the SSH model. **b** Compressed linear response method quantum circuit run on *ibm_auckland* (For system and calibration details, see Supplementary Note 5). *XY* indicates a rotation about *XX* followed by *YY*, or to be more explicit, it is $\exp(i\theta(XX + YY)) = \exp(i\theta XX)\exp(i\theta YY)$. We find the angles of each gate by the algebraic compression method given in[50,51]. **c** Fermionic correlation function $\mathcal{L}_k(t) = 2\,\mathrm{Re}\,G_k(t)$ for $\delta = 0$ using the commutator method. Data for

other values of $\delta$ are available in the Supplementary Note 1. **d** Normalized power spectrum $|\mathcal{L}_k(\omega)|^2$. **e–g** Normalized false-color plots of $|\mathcal{L}_k(\omega)|^2$ for $\delta = \{0, 0.4, 0.8\}$. Green dashed lines indicate the expected bounds of the gap, and the red lines the analytically obtained spectrum. **h, i** Normalized false-color plot of post-selected $\langle \Phi_0^y | \Phi_0^y \rangle$ and $\langle \Phi_1^y | \Phi_1^y \rangle$, respectively (see text for definition). The projected norms contain the same spectral information as $\mathcal{L}_k(\omega)$.

used to apply $e^{i\eta\mathbf{B}}$, and $\alpha_{\mathrm{comm}}^1$ is defined in ref. 47 Theorem 1. Finally, the statistical error of measurement is $\mathcal{O}(||\mathbf{A}||^2||\mathbf{B}||^2/\epsilon_{\mathrm{NL}}\sqrt{N_{\mathrm{shot}}})$, where $N_{\mathrm{shot}}$ is the sample size, or number of shots.

If the operators $\mathbf{A}$ and $\mathbf{B}$ are linear combinations of Poly(n) Pauli strings where $n$ is the number of qubits or the system size, and assuming the coefficients do not depend on $n$, we find that the spectral norms $||\mathbf{A}||$, $||\mathbf{B}||$ and the commutative norm $\alpha_{\mathrm{comm}}^1(\mathbf{B})$ scale polynomially with the system size, which lead to a requirement of $N_{\mathrm{shot}} = \mathcal{O}(\mathrm{Poly}(n)/\epsilon_{\mathrm{NL}}^2\epsilon_{\mathrm{meas}}^2)$ number of shots to keep the measurement error less than $\epsilon_{\mathrm{meas}}$. Thus, the method is scalable for various cases including momentum definite response functions.

For many response functions the operators $\mathbf{A}$ and $\mathbf{B}$ are intensive quantities, and thus are normalized accordingly with the system size. In turn, their operator norm and hence the response function's maximum possible value is 1. When $\alpha_{\mathrm{comm}}^1(\mathbf{B}) = 0$ the Poly(n) term vanishes, and $N_{\mathrm{shot}} = \mathcal{O}(1/\epsilon_{\mathrm{NL}}^2\epsilon_{\mathrm{meas}}^2)$, completely independent of the system size. For spin systems where excitations of the response function are linear combinations of commuting local variables, this leads to a better result compared to the Hadamard test, which has a quadratically scaling number of shots with the system size.

However for response functions such as momentum-definite single particle fermion Green's functions, because none of the Pauli strings in $\mathbf{B}$ commute with each other, $\alpha_{\mathrm{comm}}^1(\mathbf{B}) = \mathcal{O}(n)$, and $\epsilon_{\mathrm{NL}} = \mathcal{O}(r/n)$. In Supplementary Note 7, and as discussed in ref. 48, we observe that the theoretical error bounds are loose and lead to higher resource estimates than required. As such, even $r = 1$ might be sufficient to achieve good results, although the theoretically estimated number of shots can also be made independent of the system size by choosing the number of Trotter steps $r = n$.

### Green's function of the SSH model
We demonstrate the linear response approach by calculating the fermionic Green's function as would be measured by ARPES (angle-

resolved photoemission spectroscopy). We study a minimal model for a charge density wave known as the Su–Schrieffer–Heeger (SSH) model – an N-site 1D spinless free fermionic chain with nearest-neighbor bond-dependent hoppings (see Fig. 3a) – in the limit where the lattice distortion is static,

$$\mathcal{H}_0 = -\sum_{\langle i,j \rangle}\left[V_{nn} + (-1)^i\delta/2\right]c_i^\dagger c_j - \mu\sum_i c_i^\dagger c_i. \quad (20)$$

For finite $\delta$ this model exhibits a charge density wave, with a gap proportional to $\delta$.

To reduce our quantum computing resource needs, we apply a number of simplifications particular to this example. First, we address the issue of ground state preparation. Normally to calculate the single particle Green's function for this model, one should first generate the ground state of the model. There are numerous methods to generate the ground state or an approximate ground state for a given model[34]. The Green's function then can be calculated by our linear response method. Since our purpose here is to demonstrate our algorithm, we adjust the chemical potential to $\mu = -5$ so that the ground state is the trivial no particle state. To be clear, our method can calculate the response function on any state. But for that response function to be a Green's function, the state must be the ground state.

The second simplification is made in the operators we measure and apply, i.e. $\mathbf{A}$ and $\mathbf{B}$. We use a momentum-selective instantaneous (and thus broadband) driving field coupled to the particle creation and annihilation operators that act on all the sites $i$,

$$\mathbf{B} = \sum_i \cos(kr_i)\left[c_i + c_i^\dagger\right], \quad (21)$$

with a pulse $h(t) = \eta\delta(t)$, where we used $\eta = 0.04$. Because the ground state contains no particles, this operator is equivalent to

$\mathbf{B} \equiv \sum_i \cos(kr_i) X_i$, which is much simpler to implement, because all Pauli strings within it are 1-qubit operators, and commute with each other, therefore avoiding Trotter error. We measure $\mathbf{A} = X_0 = c_0 + c_0^\dagger$ which is local in position, and includes all momentum modes. Due to momentum conservation, measuring any $\tilde{X}_i$ would lead to the same result, but we choose $\mathbf{A} = \tilde{X}_0 = X_0$ to minimize the weight of the measured Pauli string (and thus reduce measurement noise[49]). By measuring $X_0$ we obtain

$$\mathcal{L}_k(t) = -i\langle 0| \left[ X_0(t), \sum_i \cos(kr_i) X_i \right] |0\rangle. \tag{22}$$

In the frequency basis, this becomes (see Supplementary Note 4 for details)

$$\mathcal{L}_k(\omega) = G_k(\omega) + G_k(-\omega)^*. \tag{23}$$

In general, to measure $G_k(\omega)$, one should isolate the spectrum of $\mathcal{L}_k(\omega)$ from $G_k(-\omega)^*$, which requires measuring $\mathbf{A} = Y_0$ as well. However, since $\mu = -5$, for this model the single particle energies are manifestly positive, and the interference between $G_k(\omega)$ and $G_k(-\omega)$ is negligible. Thus, $|\mathcal{L}_k(\omega)|^2$ tracks the quasi-particle peaks in $\mathrm{Im}\, G_k(\omega)$, and measuring $\mathcal{L}_k(\omega)$ is sufficient to obtain the single-particle spectrum.

We performed the calculation on the 27-qubit *ibm_auckland* superconducting quantum computer (for system and calibration details, see Supplementary Note 5) for an $N = 8$-site chain, which has allowed momentum values $k = \frac{2\pi}{N} j, j \in \{0 \ldots 7\}$. Since the driving field **B** is symmetric in $k$, both $k$ and $-k$ are obtained at the same time. We used a compressed free fermionic evolution, which is discussed in detail in refs. 50,51 and further simplified into the circuit in Fig. 3b (see Supplementary Note 4 for details). Figure 3c shows the raw data for $\mathcal{L}_k(t)$ with $\delta = 0$ at each unique $k$; the data was obtained from *ibm_auckland* via the parity operator method. The power spectrum is shown in Fig. 3d–e. While the data from the quantum computer appears quite noisy, in the frequency regime of interest there is only a single peak present in the Fourier transform, illustrating the remarkable strength of a momentum-selective probe, which picks out the single energy at each momentum, together with Fourier filtering. Upon increasing $\delta$ (Fig. 3f, g), a gap opens up in the spectrum (time traces and Fourier power spectra are available in the supplementary material). The spectrum for $\delta = 0.4$ is noisier than the other two, which we attribute to machine noise from those particular measurements. In panels h and i, we plot the norms of 0- and 1- particle components of the state right before the measurement, i.e. $\langle \Phi_0^y | \Phi_0^y \rangle$ and $\langle \Phi_1^y | \Phi_1^y \rangle$, where $|\Phi_M^y\rangle$ is defined above Eq. (15). Both methods faithfully reproduce the power spectrum, with slightly higher levels of noise for post-selection on $N = 1$.

For each $k$ and $\delta$ shown in Fig. 3 we collected 3 data sets with 8000 shots each, yielding 24,000 shots total per curve. We did not use read out error mitigation, however we incorporated dynamical decoupling and Pauli twirling as implemented in the *qiskit_research* package[52]. The raw and calibration data are shown in the supplementary material.

In order to further underscore the power of the momentum-selective linear response approach, we compare its effectiveness to a position-selective linear response and Hadamard test methods in Fig. 4 on a noisy simulator with one/two qubit noise of 1% and 10%, respectively. Compared to the momentum-selective linear response method, the position-selective one is noisier, but without particular structure. The Hadamard test, on the other hand, exhibits streaks that arise from leakage of signal from one momentum to the others. There are two key reasons for the differences seen in the figure. First, both position-selective and Hadamard test methods involve excitations at each position ($X_i$ in the figure). These must be combined in the post-processing with a Fourier transform. But, because a Fourier transform

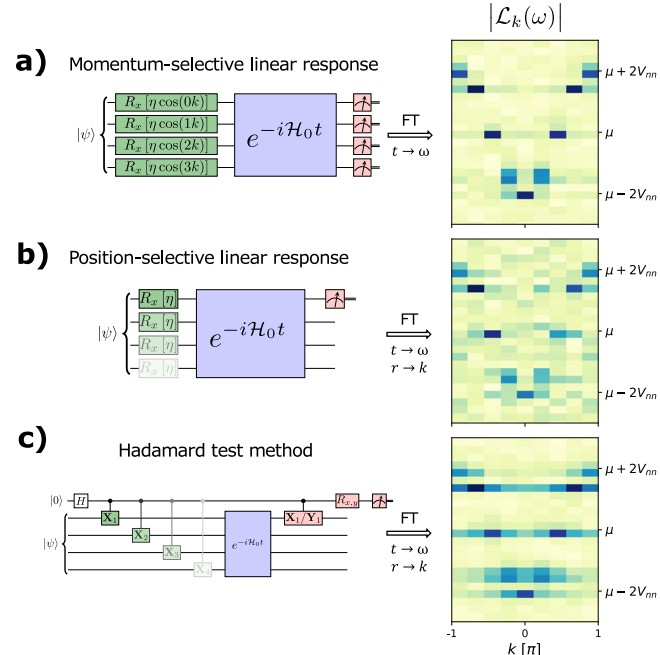

**Fig. 4 | Comparison of the momentum selective linear response, position selective linear response, and Hadamard test methods.** The circuit diagrams schematically represent the 3 approaches: **a** linear response with momentum selectivity, **b** linear response in position space and **c** Hadamard test in position space. The simulations were run on a noisy simulators with one/two qubit noise of 1% and 10%, respectively. While the momentum selective linear response method directly yields $\mathcal{L}_k(t)$, an additional spatial Fourier transformation is needed for the other two methods.

relies on constructive/destructive interference between signals, and we are performing this on noisy data, the interference is not perfect, which leads to leakage between momentum channels. When the circuits to be run for each $X_i$ are not identical in structure, the noise becomes dependent on $i$. This also leads to a momentum-dependent noise term $f_k$, which in turn appears in the momentum space signal as

$$G_k^{\mathrm{observed}}(t) \propto \sum_p f_{k-p} G_p^{\mathrm{exact}}(t), \tag{24}$$

which we derive in the Supplementary Note 2. Second, the Hadamard test method introduces more of the same problem because each $X_i$ is a separate circuit — in addition to needing more circuits to be run and an additional ancilla. This further exacerbates the issue with the Fourier analysis. The momentum-selectivity avoids these issues by making a unique excitation and thus producing a response function with a single large contribution.

### Green's function of the 1D Hubbard model
The method is equally applicable to interacting models that are not simply integrable; here we demonstrate this by calculating the Green's function of a periodic 1D Hubbard model with $N = 6$ sites (12 spin-orbitals). The Hamiltonian for the model is

$$\mathcal{H} = -t \sum_{\langle i,j\rangle,\sigma} \left( c_{i,\sigma}^\dagger c_{j,\sigma} + c_{j,\sigma}^\dagger c_{i,\sigma} \right) + U \sum_i n_{i,\uparrow} n_{i,\downarrow} + \mu \sum_{i,\sigma} n_{i,\sigma} \tag{25}$$

where $n_{i,\sigma} = c_{i,\sigma}^\dagger c_{i,\sigma}$. The first term in the Hamiltonian is the nearest-neighbor hopping between sites, the second term governs the on-site electron-electron interactions with strength $U$, and the last term sets the chemical potential for the model. We set $\mu = 0$, so the ground state of the system is half-filled. The calculations were performed using a

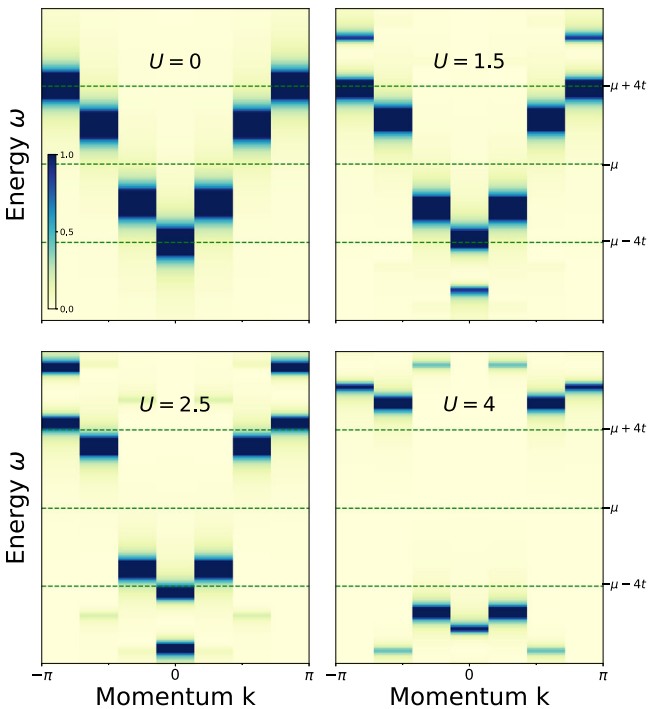

**Fig. 5 | Green functions for the 1D Hubbard Model.** False color plot of $|G_k^R(\omega)|^2$ for various values of $U$. The interactions add a shift proportional to the magnitude of $U$ to each of the excitation peaks in the spectrum and additional satellite peaks start to appear.

statevector simulator, with the ground state prepared via exact diagonalization. We obtained the retarded Green's function via Eq. (10) for different interaction strengths $U$ using a momentum-selective driving field (c.f. Eq. (21)) and a Gaussian temporal profile. The results are shown in Fig. 5 for various values of $U$. With increasing interaction strength, a gap opens in the spectrum, satellite peaks appear, and a reduction in the weights of the sub-bands occurs, all in agreement with the known results for this model[53,54].

### Polarizability of the SSH model

We next consider the polarizability $\chi(q,\omega)$ of the 1D chain. The polarizability is the response of the electronic system to an applied potential. It plays a critical role in the screening of interactions between electrons in solids and molecules, and in their electromagnetic properties. Experimentally, the polarizability can be studied by light absorption or scattering, or by momentum-resolved electron energy loss spectroscopy (M-EELS). The polarizability is defined by

$$\chi(r,t) = -i\langle\psi_0|\delta n(r,t)\delta n(r=0,t=0)|\psi_0\rangle, \quad (26)$$

i.e. it is a charge-charge correlation function. Here $\delta n$ is the change in the charge from the equilibrium density. The observable **A** is the charge, and the applied field **B** (which is coupled to the charge) is a potential. The excitations are changes in the density, which are composed of pairs of fermionic operators, and thus this is a bosonic correlation function.

For this demonstration, **B** acts on a single site, and we classically simulate a partially filled 24-site chain ($\mu = 0.9$, which was chosen to reveal the salient features of a 1D system). As discussed above, one of the advantages of the linear response framework is that all 24 correlation functions are obtained with a single calculation. Figure 6a shows $\text{Im}\chi(q,\omega)$, which is the double Fourier transform of $\chi(r,t)$ obtained from driving a single site with a sharp $h(t)$. $\text{Im}\chi(q,\omega)$ has all the text-book features of the response of a 1D charged system[55]; there is no

response at all at $q = 0$ due to charge conservation, there is a narrow dispersive feature at low $q,\omega$ that broadens with increasing $q$, and a low-energy turnover with a minimum at $2k_F$.

Since $h(\omega)$ has support across the entire spectrum of $\chi(q,\omega)$ (shown in the inset), the entire spectrum can be obtained from this measurement. This is in contrast to panel b, where we drive with a short-duration sinusoid centered at $\omega = 1.5$. This excitation is frequency-selective; that is, it only excites the system at frequencies where $h(\omega)$ has finite support. This range of frequencies is indicated by dashed lines in the figure. With our particular choice of $h(t)$, we are able to observe some of the middle range of excitations, but are insensitive to the lower frequencies and the top of the spectrum. Note that there is no restriction on the Fourier transform of $\chi(r,t)$ per se; rather, the need to divide by $h(\omega)$ (see Eq. (4)) limits the applicable window to the ranges where $h(\omega)$ is sufficiently far from zero.

### Discussion

The linear-response based formalism is a shift in perspective on quantum simulation where the experiment itself is simulated. This is in contrast to Hadamard test based approaches, where the system is simulated and the desired observables are extracted either outside of the system qubits and/or from a large excitation. This shift in perspective and methodology enables a much broader set of observables to be envisioned and easily calculated without additional post-processing. And, unlike variational methods for computing response functions[18–22], it relies almost entirely on time evolution, a task for which quantum computers are well-suited.

The linear response method enables efficiently obtaining a number of dynamical properties. The fact that it is efficient is important because we expect quantum computation to remain costly for quite some time – this is true on today's NISQ hardware, and will also be true for early fault tolerant quantum computing. Thus, if the properties can be obtained in as few circuits as possible, and with a higher tolerance for error, this is beneficial. Our proposed algorithm achieves precisely this.

This shift in perspective and the resulting implementation leads to several clear advantages:

1.  Efficiency is achieved by enabling the measurement of many correlation functions at the same time (see also Gustafson et al.[35]). For example, a local excitation can provide a researcher access to $\chi(r,t)$ given in (26) for all values of $r$ because all $n_r$ can be measured simultaneously.
2.  Additional efficiency and noise resilience are achieved by enabling the excitation/measurement to be any Hermitian operator, potentially obviating the need to run many circuits and having to post-process the data from multiple different circuits, each with its own noise characteristic.
3.  Being able to make tailored excitations means that researchers can pinpoint the regime they are interested in and study precisely the excitations of interest, within a single circuit. This is an improvement over rather than having to extract the signal of interest out of the full system response (plus the noise), or having to rely on cancellation between the (noisy) results obtained from several circuit runs.
4.  Fault tolerant quantum computing comes at a significant qubit overhead cost. An algorithm that has a high tolerance for error and requires 1 fewer qubit will enable earlier hardware calculation of response functions.

Fermionic response functions (anti-correlation functions) can be obtained with the same experimentally centered, linear response perspective; this is unlike other ancilla-free methods[35,36] which are limited to bosonic response functions. The post-selection method is intuitive, as the particle number sectors are clearly delineated. On the other hand, the auxiliary operator method is an unusual perspective; it

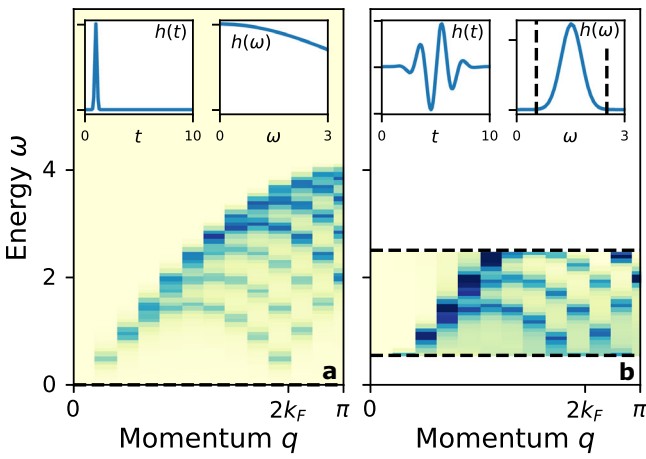

**Fig. 6 | Polarizability for the 1D chain.** Both panels show $\mathrm{Im}\,\chi(q,\omega)$ in false color. The insets show the driving field $h(t)$ and its Fourier transform. **a** $\chi(q,\omega)$ obtained from the response due to a sharp excitation with height 0.1. **b** $\chi(q,\omega)$ obtained from the response of a frequency selective field. The dashed lines indicate the range where $|h(\omega)|^2 < 10^{-3}$. Here, we used $h(t)$ as sinusoid with a Gaussian profile of width $\sigma = 0.625$, height 0.05, and centered at $\omega = 1.5$.

is sufficient to measure almost the same operator as for the bosonic correlation function. The electron Green's function, for example, is obtained simply by keeping track of the parity as well as the occupation number measurement. In either case, this is an important advance since electron Green's functions play a key role in physics; as an important measurement per se[56], and as an ingredient in embedding theories such as dynamical mean field theory[3–8].

While here we have explicitly demonstrated the linear response approach in the context of a charge density wave, it is a general method to obtain response functions, and is not limited to electronic Hamiltonians. It can be applied to spin or bosonic models, or other models from fields where quantum simulation plays a role, including chemistry and high energy physics. Different choices of **A** and **B** extend the method to a wide variety of observables. For example, the conductivity is a current-current correlation function, for which $h(t)$ is an applied electric field. A $zz$-spin susceptibility can be obtained with $h(t)$ as a $z$-axis magnetic field, and the operators $\mathbf{A} = \mathbf{B} = S_z$. Moving forward, the functional derivative formalism can be extended to higher order derivatives that involve multiple driving fields. One notable application is resonant inelastic X-ray scattering (RIXS), which is a four-point correlation function[57], which is very challenging to calculate via diagrammatics. In addition, and aside from direct experimental probes, pairing vertices in superconductors and other ordered phenomena also fall into this class of observables. We reserve these discussions for future work.

## Data availability
All data needed to evaluate the conclusions in the paper are present in the paper and/or the Supplementary Materials. The data for the Figs. 3, 4, 5 and 6 are available at https://doi.org/10.5061/dryad.51c59zwcn.

## Code availability
Code is available from the corresponding author upon request.

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

## Acknowledgements

We acknowledge helpful discussions with Erik Gustafson and Vito Scarola. We acknowledge the use of IBM Q via the IBM Q Hub at North Carolina (NC) State for this paper. The views expressed are those of the authors and do not reflect the official policy or position of the IBM Q Hub at NC State, IBM or the IBM Q team. We acknowledge the use of the QISKIT software package[52] for performing the quantum simulations. This work was supported by the Department of Energy, Office of Basic Energy Sciences, Division of Materials Sciences and Engineering under grant no. DE-SC0023231. J.K.F. was also supported by the McDevitt bequest at Georgetown.

## Author contributions

A.F.K. and J.K.F. conceptualized the project. A.F.K. developed the methodology, performed the quantum computer experiments, and ran the polarizability calculations. E.K. contributed to the mathematical development for fermionic response functions, error bound calculations and designed the quantum circuits. H.A.L. ran the noisy quantum simulator and Hubbard model calculations. All authors discussed the results and contributed to the development of the manuscript.

## Competing interests

The authors declare no competing interests.
