## [Peer Review File · Nature Communications]

REVIEWER COMMENTS

Reviewer #1 (Remarks to the Author):

The authors outline a linear response-based method to evaluate bosonic and fermionic correlation functions. Frequency and momentum selectivity can be achieved by choosing the frequency support or setting the momentum of the perturbing field. Auxiliary operator method and post-selection method are introduced to facilitate the real-time approach for fermionic correlation function calculation. Critical technical details like damping function and practical procedures are clearly described. The demonstrations on simulator and hardware are nevertheless not state-of-the-art, as the model is noninteracting and therefore not hard classically. The fixed-depth compressed free fermionic evolution is also restricted to non-interacting system. With that being said, they serve the purpose for illustrating the outlined quantum-inspired method. The manuscript generally reads well, except for a few places to clarify or elaborate. I would recommend the acceptance of the manuscript, given that the authors satisfactorily address the following points.

- 1) Can the authors have some discussion on the scalability and potential practical quantum advantage of the outlined method? The 0th step of the method is to prepare the ground state of the quantum system, and recently there is some discussion on lack of evidence of exponential quantum advantage of ground state preparation ([1] S. Lee et al., Evaluating the Evidence for Exponential Quantum Advantage in Ground-State Quantum Chemistry, Nat Commun 14, 1 (2023).).
- 2) In the discussion of advantages of the method in the beginning paragraph of page 2, can the authors clarify what other methods it is compared to? For instance, I think the ancilla-based approach can also deal with generic operators.
- 3) The definition of \tilde{X}_r near the end of page 2 in the momentum selectivity paragraph is absurd, because the Jordan-Wigner encoding has not been mentioned yet.
- 4) Can the authors elaborate on the diagonal form of the response function in momentum space? Specifically, does it apply to both non-interacting and interacting periodic systems?
- 5) In the discussion of auxiliary operator method, it is not clear how it reaches the point of replacing A with $X_i P$ and B with X_j . The general recipe is to replace $A(t)$ with $A(t)P(t)$. And for the specific case, $A(t) = \delta(t) c_i$, which has a tail of Z's. It is not clear how that simple replaced form of A and B are derived.
- 6) In the paragraph above "RESULTS" section, the authors have statements like "This work also has impact on classical computing via a quantum inspired algorithm. The approach described below allows for one to compute response functions by simply running time evolution on a classical computer. This provides a different paradigm for computing ...". It should probably be toned down. For instance, it has been practiced to use real time evolution to get response function in the (time-dependent) DMRG community.

7) In the caption of Fig.2, could the authors clarify whether the XY gate is to be applied as XX gate followed by YY gate, or it refers to some rotation with XX and YY as the generator?

8) It is not clear what “so we set $\mu = 5$ to suppress the initial total electron number”? Is it a canonical or grand canonical system? What is the filling of the model and is it fixed?

9) Please clarify whether readout error mitigation has been applied.

10) The Fourier amplitudes are also important, and it is mentioned in the text that that they are provided in supplementary materials. But they are missing.

11) There is discussion about 8000 shots each on page 6. Can the authors clarify how many measurement circuits are there? I think it is not just one, as there are combinations of X's and Y's.

12) In the simulation of polarizability, why is μ changed to 0.9?

13) In the final discussion of “B can be chosen to be non-unitary because we apply $e^{-ih(t)B\Delta t}$, as opposed to the Hadamard test which applies B on the state.” I assume any operator can be expanded in terms of a sum of pauli strings, and correlation function can be always obtained as a sum of individual terms using Hadamard test. Can the authors clarify the issue here?

Other minor points:

1) Please specify the full t-interval to be integrated over in Eq. (1b).

2) Fix the typo for A(t) after Eq. (3).

3) It would be good to spell out explicitly that the SSH model is a spinless fermionic model.

4) Is there a typo in “the driving field is implemented ain a single Trotter step”?

5) Panel b is referred is mentioned before mentioning fig. 2.

Reviewer #2 (Remarks to the Author):

In this work, the authors present a framework to perform linear response calculations for model Hamiltonians on quantum computers. The proposed solution combines basic time-evolution subroutines with experiment-mimicking procedures to perturb the system and observe dynamical correlations. This solution is presented as an alternative to other well-known protocols such as the Hadamard test, and according to the authors it offers distinctive advantages, including an intuitive way to enforce momentum and frequency selectivity as well as an increased resilience to hardware noise.

The scientific work is sound, and the topic is certainly of interest for the community of researchers interested in digital quantum simulations. These ‘spectroscopic’ calculations have been extensively studied in the literature and are considered among the most promising use-cases for current and future quantum processors.

However, I am not convinced that this work, in its present form, meets the criteria of novelty and impact for Nature Communications. The basic idea is essentially an application of textbook definitions: while simplicity is not per se a flaw (actually, most of the times the opposite is true), one would need stronger arguments to convince the reader that the proposed solution has the potential to critically surpass current state-of-the-art methodologies.

A direct way for doing so would be to assess the feasibility of, and ideally perform, larger scale implementations of the algorithms on quantum processors, targeting a more general class of models that – at least in some limit – can become classically hard to simulate. Discussing in concrete terms whether the proposed method may lead to experiments in the realm of few tens to hundreds of qubits, where competing classical and quantum methods could fail or require substantially larger efforts, would represent a key addition to the current study.

More in general, I believe that the analysis presented by the authors is potentially interesting, but essentially incomplete. Some specific points are listed below:

- An analysis of the digital unitary decomposition (Trotterization) error is missing. This is an important point to consider if one must implement general time-dependent evolution terms. Overall, it would be important to see concrete scaling arguments concerning the complexity of quantum circuits (and quantum measurements) necessary to obtain faithful results as a function of the desired frequency and momentum resolution and/or support, and of course as a function of system size.
- For the benefit of the readers, a more explicit, step-by-step derivation of the quantum circuits used in the specific experiments, and their compilation on quantum hardware, could also be helpful.
- The two alternative approaches proposed for fermionic correlations are potentially interesting. However, both methods seem applicable only to certain restricted classes of Hamiltonians: it would be nice to extend the discussion on this point, and for example clearly identify classes of hard problems that the methods could tackle. Moreover, for the auxiliary operator method the authors propose the use of a parity operator, which has a very non-local support: these observables are usually more challenging to measure in the presence of noise – see e.g., Nature Physics 19, 752 (2023) –, as also mentioned by the authors on Page 6. Can this become a limitation to the scalability and noise resilience of the proposed protocol?
- Concerning the post-selection method, I find the notation in Eqs. 12 not very clear. Probably, moving some additional material from the SI to the main text could be helpful. Are some of these

quantities non-normalized wavefunction components? How, and (most importantly) how efficiently, can these be reconstructed from measurements taken on a quantum computer? In general, how do the sampling costs of the post-selection scale with the system size?

- On page 4, the statement “This can be tested by repeating the simulation [...] and checking that the response scales similarly” sounds very heuristic. Can anything more systematic be said? For larger problem instances, it could become crucial to be able to specify the experimental details beforehand.

- On page 5, “Since $\mu=5$, for this model the single particle energies are manifestly positive”: how is the procedure generalized when such knowledge is not available?

- On page 6, “To minimize the weight of the measured Pauli string (and thus reduce measurement noise) we perform the measurement on the 1st qubit”. Could this be a limitation in general (e.g., reveal boundary effects)?

- Page 6, is there any practical explanation for the results to be significantly noisier for $\delta=0.4$? Are for instance the quantum circuits more complex?

- In all experimental results on the SSH model, I think a clear comparison (e.g., in the plots) with the expected, exact results is missing.

- The discussion at the end of the section “Green’s function of the SSH model”, concerning Fourier analysis and the impact of noise, is potentially interesting and could become a strong point in favor of the proposed method. Can it be expanded and supported with some more formal studies?

- I do not really understand the focus (starting from the title) on the “quantum-inspired” aspect of the method, namely the attempt to propose this also as an alternative solution for classical simulations. One could have the impression that the authors themselves have doubts on the scalability and applicability of the method on quantum computers. The idea of simulating time evolution on classical computers, as opposed to, e.g., approaches based on the Lehmann formula, is in fact quite natural: in practice, however, time evolution is usually hard to realize on classical computers. The authors should support their statements with (1) a more extensive survey in the classical state-of-the-art, identifying the challenges that their method could solve (2) some more concrete computational cost and scaling arguments in the classical domain (corroborating very generic statements like “it is likely to be much more efficient than currently used methods.”

- Overall, I believe the quality of the exposition could be largely improved, particularly to make the whole discussion more accessible to a broader audience. A few examples:

- On Page 2, for instance, a statement like “using the convolution theorem” could be accompanied with a reference, or the definition of X_t (part of B) seems to implicitly assume a Jordan-Wigner mapping, which might confuse non-experts.

- Competing approaches (e.g., the Hadamard test) are never really defined along the discussion.

- The final paragraph of the “Bosonic (commutator) correlation functions” is very hard to follow. What are the “other approaches” or the embedding techniques the authors refer to?

- “Superconducting terms” (page 4) might not tell much to non-specialists.

- Page 6, typo “ain a single Trotter step”.
- Page 6, “ibm_auckland”: the authors could say something more about the processor (technology, provider, number of qubits, topology, etc...) -- not every reader might recognize the name.
- Page 6, “form of the quantum circuit shown in panel b”: I think the authors mean Fig 2, but a precise reference is missing.
- Page 7, the sentence “This is enabled on the quantum circuit level...as opposed to the Hadamard test which applies B on the state” is a bit obscure.

Reviewer #3 (Remarks to the Author):

In their manuscript, Kökcü et al. introduce a method to retrieve linear response functions from a quantum simulator. Unlike other existing methods that rely on the computation of correlation functions, here the authors propose to implement in the quantum simulation of a many-body system an actual perturbation to the dynamics and then measure the response of the system in the signal of an operator of interest. The correlation function is then retrieved from the measured signal by taking a functional derivative w.r.t. the performed perturbation. According to the authors, this represents a shift in paradigm that brings a number of benefits w.r.t. to existing methods. For example, it allows for frequency and momentum selectivity, which can lead to a reduction in circuit noise for a number of relevant cases, additionally, the method is ancilla free and it can be used to compute correlation functions of non-unitary operators. To illustrate their method, the authors compute Green's functions of a charge density wave based on the Su-Schrieffer-Heger model using the ibm_auckland quantum computer.

In general, the manuscript is well-written and clear, and the derivations are sound and support the performed analysis. I am, however, unsure of the significance of the result and fail to see the shift of paradigm that the authors claim. To justify the relevance of the introduced framework the authors state a number of ambiguous statements that, in my view, fail to convincingly show the disruptive character of their method. For example, the authors say that

“The measurement process is truly a part of the simulation as an experimental driving field. This is in contrast to [...] other competing approaches, where the simulation is limited to the system, and the desired observables are extracted either outside of the system qubits and/or from a large excitation.”

It is unclear to me what is meant here by “the measurement process is part of the system” or by “experimental driving field”. Similarly, what is it meant by “observables are extracted outside of the system”? Or why is this worse than the approach here?

The authors also say that “this shift in perspective and methodology enables a much broader set of observables to be envisioned and easily calculated, and enables a direct connection to experiment” What is a direct connection to experiment? Why are these set of observables easier to calculate? These claims need to be clarified and supported with concrete evidence to establish the significance of the proposed method.

Additionally, the authors state that “it relies almost entirely on time evolution, a task for which quantum computers are naturally suited.” Again, this sounds very ambiguous and fails to show with precision what advantage exactly this brings. Are other approaches not based on time evolution? Are they based on some task that quantum computers are not good at?

The fact that the protocol does not make use of an ancillary system is also stated as an advantage of the method, as it does not require maintaining coherence between the ancilla and the system. This, however, does not seem a significant difference in practice as the current method still requires maintaining coherence between the N qubits of the system, which is just one less qubit than ancilla-based methods that require maintaining coherence between $N+1$ qubits.

The list of hand-waving statements that attempt to convince the reader of a radical superiority of the introduced method w.r.t. to existing methods goes on, but I think my point is clear. The manuscript lacks a comprehensive comparative analysis with existing methods for retrieving linear response functions. The authors make bold claims about the advantages of their approach but do not provide a clear comparison with other approaches based on different principles. The advantages of the method over existing techniques need to be better demonstrated and supported with specific examples to validate the claims.

Other minor comments:

In the time-dependent expression for $A(t)$ after Eq. (3), a minus sign is missing in the exponential

In the definition of B before Eq. (5) a factor of 2 is missing in front of the cosine after the second equality

In conclusion, I find the manuscript's proposed method for retrieving linear response functions in quantum simulators to be interesting and well-executed. However, the significance of the results and the claimed shift in paradigm remain unclear due to the lack of concrete evidence and comparative analysis with existing methods. While this is probably a valuable addition to the toolbox of quantum simulation techniques for response theory, I believe, it lacks the significance required for publication in Nat. Commun. I, thus, recommend addressing the concerns raised above and submitting the work to a more specialized journal where the potential value of this method can be better appreciated within a more focused scientific community.

REVIEWER COMMENTS

Reviewer #1 (Remarks to the Author):

The authors outline a linear response-based method to evaluate bosonic and fermionic correlation functions. Frequency and momentum selectivity can be achieved by choosing the frequency support or setting the momentum of the perturbing field. Auxiliary operator method and post-selection method are introduced to facilitate the real-time approach for fermionic correlation function calculation. Critical technical details like damping function and practical procedures are clearly described. The demonstrations on simulator and hardware are nevertheless not state-of-the art, as the model is noninteracting and therefore not hard classically. The fixed-depth compressed free fermionic evolution is also restricted to non-interacting system. With that being said, they serve the purpose for illustrating the outlined quantum-inspired method. The manuscript generally reads well, except for a few places to clarify or elaborate. I would recommend the acceptance of the manuscript, given that the authors satisfactorily address the following points.

We thank the reviewer for their careful reading of the manuscript, and their positive assessment. Below, we address the points raised by the reviewer one by one. Moreover, we have added a demonstration on a simulator of an interacting model, Hubbard Model, illustrating that the method works here as well.

1) Can the authors have some discussion on the scalability and potential practical quantum advantage of the outlined method? The 0th step of the method is to prepare the ground state of the quantum system, and recently there is some discussion on lack of evidence of exponential quantum advantage of ground state preparation ([1] S. Lee et al., Evaluating the Evidence for Exponential Quantum Advantage in Ground-State Quantum Chemistry, Nat Commun 14, 1 (2023).).

The reviewer makes an excellent observation about the field. We agree that preparing the ground state (the 0th step) is indeed a complex question, which is an entire field of research in its own right. The paper mentioned by the reviewer is an indication of the active debate in the field. Our work answers the question of what to do with a ground state once one is obtained by any of the methods being used and developed in the field. These could include variational algorithms such as VQE, but also subspace methods, adiabatic or imaginary time evolution, and beyond. Our linear response approach is agnostic to the method used to obtain the ground state. Thus, we focus on the various aspects of the approach in that light.

Moreover, we would like to point out that our method is not restricted to being used for ground states. In fact, any pure or mixed state can lead to a response, and the linear response method can be used independently of what the input state is. Furthermore, even if the ground state is created efficiently enough to continue to a response function calculation, often the complexity of the latter could still make the problem too difficult to solve. By making the response function calculations efficient, we enable a novel way to determine them. Incorporating vertex corrections

for complex response functions, such as resonant inelastic x-ray scattering, is currently only possible by direct measurement approaches (such as exact diagonalization and time-evolution). Our work provides a new alternative.

We have amended the manuscript in the Results section, where we briefly outline these issues and provide a reference to a summary that includes discussion of algorithm complexities.

2) In the discussion of advantages of the method in the beginning paragraph of page 2, can the authors clarify what other methods it is compared to? For instance, I think the ancilla-based approach can also deal with generic operators.

We thank the reviewer for pointing out that the comparisons were unclear. We have amended the manuscript to provide a more clear comparison in the “State of the art: Hadamard test” section.

As the reviewer correctly indicates, the ancilla based method is capable of dealing with generic operators as well. Let us consider the momentum creation operator for instance. In the position space, the operator consists of $O(n)$ single qubit operators, where n is the number of qubits/system size. Implementation of this with the Hadamard test can be done by running each of those single qubit operators separately, and then adding the results in postprocessing. To do so, one needs to run the circuit $O(n)$ times. However, our algorithm requires us to append only $O(n)$ single qubit gates into one circuit, and run that one circuit. Thus, even though the Hadamard test method can handle generic operators in an indirect way, our method can handle it in a direct way with fewer of circuit runs. And, as we detail further, the postprocessing leads to additional errors.

A second way of implementing momentum definite response in the Hadamard test, which does only use one circuit, is via Linear Combinations of Unitaries (LCU) by A. Childs 2012. The method applies an operator A on a given state regardless if A is a unitary or not. To do this, one must use $\log(n)$ ancilla qubits and append $O(n)$ 2-qubit operations, which is costly by itself. In addition, LCU is probabilistic if A is not a unitary operator, and only a fraction of the circuit runs produce valuable results if the targeted operator. If A is unitary, the success rate can be increased to 100% with the application of Oblivious Amplitude Amplification (OAA) D.W.Berry 2014, at the cost of a substantial increase in the circuit depth, which makes it not applicable to noisy quantum hardware.

We have amended the manuscript to make the points above.

3) The definition of \tilde{X}_r near the end of page 2 in the momentum selectivity paragraph is absurd, because the Jordan-Wigner encoding has not been mentioned yet.

We have replaced it with the magnon creation operator with definite momentum to avoid the confusion.

4) Can the authors elaborate on the diagonal form of the response function in momentum space? Specifically, does it apply to both non-interacting and interacting periodic systems?

Yes it applies. In any translationally invariant model, including the interacting fermionic models, momentum is conserved. Any response function, i.e. a two point correlation function has to be diagonal in momentum space due to momentum conservation. We have amended the text with a 2-3 sentence explanation right after Eq. 6.

5) In the discussion of auxiliary operator method, it is not clear how it reaches the point of replacing A with $X_i P$ and B with X_j . The general recipe is to replace $A(t)$ with $A(t)P(t)$. And for the specific case, $A(t) = \delta(t) c_i$, which has a tail of Z 's. It is not clear how that simple replaced form of A and B are derived.

Thank you for catching this error, there is a typographical mistake. X_i and Y_i should have had the Jordan Wigner string attached, and we amended the manuscript by replacing them with $Z_0 Z_1 \dots Z_{i-1} X_i$ and $Z_0 Z_1 \dots Z_{i-1} Y_i$.

6) In the paragraph above "RESULTS" section, the authors have statements like "This work also has impact on classical computing via a quantum inspired algorithm. The approach described below allows for one to compute response functions by simply running time evolution on a classical computer. This provides a different paradigm for computing ...". It should probably be toned down. For instance, it has been practiced to use real time evolution to get response function in the (time-dependent) DMRG community.

We appreciate the reminder of the work within the DMRG community (it is also done in the exact diagonalization community). We have softened the language in this section, and added a few appropriate references. However, we do wish to note that the potential for obtaining fermionic correlation functions in this manner is (to the best of our knowledge) not yet known in the classical computing context, which we also point out in the revised manuscript.

7) In the caption of Fig.2, could the authors clarify whether the XY gate is to be applied as XX gate followed by YY gate, or it refers to some rotation with XX and YY as the generator?

The XY gate represents $\exp(i \theta (XX + YY)) = \exp(i \theta XX) \exp(i \theta YY)$ where θ is the rotation angle and i is the imaginary unit. We can separate the exponential into two exponentials because XX and YY commute. We have amended the caption to clarify this.

8) It is not clear what “so we set $\mu = 5$ to suppress the initial total electron number”? Is it a canonical or grand canonical system? What is the filling of the model and is it fixed?

The calculation is done in the grand canonical ensemble. For the purposes of the demonstration only, i.e. in order to make the quantum computing results tractable on today’s hardware, we have started from a virtually empty ground state and computed the particle addition spectrum. For a non-interacting model, this is the same as the spectrum.

We have amended the section in order to make this point clear.

9) Please clarify whether readout error mitigation has been applied.

No read out error mitigation was used. Only dynamical decoupling and Pauli twirling were used. To further emphasize this, we added a sentence on page 7 paragraph 3 to clarify that we have not used readout error mitigation.

10) The Fourier amplitudes are also important, and it is mentioned in the text that that they are provided in supplementary materials. But they are missing.

We apologize for the confusion – by Fourier amplitudes here we had intended to mean the Fourier power spectra; we have corrected the wording in the text. The power spectra are shown in Fig S1.

11) There is discussion about 8000 shots each on page 6. Can the authors clarify how many measurement circuits are there? I think it is not just one, as there are combinations of X’s and Y’s.

We discuss in the paper, and in the supplementary material section IV.A, that we are measuring \mathcal{L}_k , which corresponds to only an XX correlator (see SI Eq. S10). SI section IV.A is a detailed description of how this XX correlation is the same as the Green’s function *for this specific model*. Moreover, since it is just a measurement of X_0 , we only need one measurement circuit per given momentum value k .

In the more generic case however, the reviewer is right. One needs to apply c_k^\dagger , i.e. $\tilde{X}_k + i\tilde{Y}_k$, requiring 2 circuits; and one needs to measure $c_0 = X_0 + iY_0$ and this requires measurement of X_0 and Y_0 separately requiring 2 circuits as well. Thus, for the most generic case, one needs 4 circuits per measurement. We have amended the main text, talking about the specific and the general cases.

We have clarified this in the text by adding the explicit definition of \mathcal{L}_k to the manuscript as well (Eq. 18), and explaining why measuring X_0 alone is sufficient.

12) In the simulation of polarizability, why is μ changed to 0.9?

The purpose of this simulation was to show an example bosonic response function. For this set of momenta, $\mu=0.9$ highlighted some of the features of the polarizability that are particularly unique for a 1D system of fermions – specifically, it makes the excitation energy going to 0 at k_F visible within the momentum range. We have added a citation to a reference work on the polarizability (Lindhard response function) by Mihaila.

13) In the final discussion of “B can be chosen to be non-unitary because we apply $e^{-ih(t)B\Delta t}$, as opposed to the Hadamard test which applies B on the state.” I assume any operator can be expanded in terms of a sum of pauli strings, and correlation function can be always obtained as a sum of individual terms using Hadamard test. Can the authors clarify the issue here?

We understand the confusion pointed out by the reviewer. It seems like we mistakenly convey that the Hadamard test has no way of handling non-unitary operators. We have clarified this issue, and amended the text accordingly.

The reviewer is right, the Hadamard test is capable of handling linear combinations of individual Pauli strings by handling each Pauli string one by one, as we also have discussed in the answer to question 2 of the reviewer. The Hadamard test can handle a unitary A,B operator in one circuit, and Pauli strings are unitary operations as well as Hermitian. However, our method is capable of handling any Hermitian operator in one circuit *without requiring it to be a unitary operator at the same time*.

In Figure 3 C, in order to compare our method to the Hadamard test, we illustrate the result of needing multiple circuits: we measure the Green’s function in position space using the Hadamard test circuit structure, and apply Fourier transformation to obtain the Green’s function in the momentum space. As it can be seen there, the Hadamard test generates a response that has all possible frequencies in the system at all momenta, whereas the linear response method does not.

We have added an analysis of this error incurred by the Hadamard test method when taking a Fourier transform of several noisy signals. The error originates from the fact that each component of the Green’s function in the position basis is obtained from a different circuit, and therefore has a different noise. Because of this, certain terms that would have canceled out exactly in a noiseless case fail to cancel out, which leads to the “frequency leakage” seen in Fig 3C.

In addition to this change (as noted above) we have made it clear that the Hadamard test circuit structure can be used for non-unitary operators in this way as well.

Other minor points:

1) Please specify the full t-interval to be integrated over in Eq. (1b).

The lower limit of the interval is not relevant to the calculation, that is why we omitted. It is a fixed time that is earlier to any other time compared to any time value such as t and t' that are included in the calculation. In principle it could be chosen as negative infinity. We amended the equation and the text accordingly by putting t_s as the lower limit representing “starting time”, and explained that $t_s < t$ and $t_s < t'$ in the text. This is because we start in an energy eigenstate, and time evolution just modifies the global phase until a field is applied.

2) Fix the typo for $A(t)$ after Eq. (3).

The reviewer is right. The equation is missing a minus sign, we corrected it to $A(t) = \exp(itH_0) A \exp(-itH_0)$.

3) It would be good to spell out explicitly that the SSH model is a spinless fermionic model.

We have added the word “spinless” to the sentence where we first introduce SSH model.

4) Is there a typo in “the driving field is implemented ain a single Trotter step”?

Thanks to the reviewer for spotting this typo. Yes, we meant “in”, and we corrected it.

5) Panel b is referred is mentioned before mentioning fig. 2.

Thanks to the reviewer for spotting this typo causing confusion. It should be Fig. 2. panel b., we corrected it.

Reviewer #2 (Remarks to the Author):

In this work, the authors present a framework to perform linear response calculations for model Hamiltonians on quantum computers. The proposed solution combines basic time-evolution subroutines with experiment-mimicking procedures to perturb the system and observe dynamical correlations. This solution is presented as an alternative to other well-known protocols such as the Hadamard test, and according to the authors it offers distinctive advantages, including an intuitive way to enforce momentum and frequency selectivity as well as an increased resilience to hardware noise.

The scientific work is sound, and the topic is certainly of interest for the community of researchers interested in digital quantum simulations. These 'spectroscopic' calculations have been extensively studied in the literature and are considered among the most promising use-cases for current and future quantum processors.

We thank the reviewer for their supportive comments. We are in agreement that spectroscopic calculations are a promising use case for NISQ era or early fault tolerant era quantum computers, and indeed our work is aimed at advancing this cause.

However, I am not convinced that this work, in its present form, meets the criteria of novelty and impact for Nature Communications. The basic idea is essentially an application of textbook definitions: while simplicity is not per se a flaw (actually, most of the times the opposite is true), one would need stronger arguments to convince the reader that the proposed solution has the potential to critically surpass current state-of-the-art methodologies.

The reviewer is correct that we have used textbook definitions of response functions. However, as is the case with many of textbook definitions, they are not automatically implementable. In this case, we have shown that this is doable — in particular, we have implemented the (otherwise entirely formal) functional derivative definition of a response function. More than that, we have developed an extension thereof that makes the functional derivative applicable to fermionic response functions *without* the need of Grassman variables, which are also not implementable. Indeed, we have two ways of doing so, both presented in the paper, and which can be used in both quantum and classical computing.

A direct way for doing so would be to assess the feasibility of, and ideally perform, larger scale implementations of the algorithms on quantum processors, targeting a more general class of models that – at least in some limit – can become classically hard to simulate. Discussing in concrete terms whether the proposed method may lead to experiments in the realm of few tens to hundreds of qubits, where competing classical and quantum methods could fail or require substantially larger efforts, would represent a key addition to the current study.

We appreciate the opportunity to elaborate on the potential of the method.

When we get to 10s or 100s of qubits, and we manage to prepare an interesting state of some form, we need to be able to measure its properties. Generically, time evolution is easier than preparing the quantum state – it is in the BQP complexity class, in contrast to state preparation which is in QMA. The method proposed here will enable researchers to efficiently obtain a number of dynamical properties. The fact that it is efficient is important because we expect quantum computation to remain costly for quite some time – this is true today on NISQ hardware, and will also be true for early fault tolerant quantum computing. Thus, if the properties can be obtained in as few circuits as possible, and with a higher tolerance for error, this is beneficial. Our proposed algorithm achieves precisely this.

To be more specific,

- Efficiency is achieved by enabling the measurement of many correlation functions at the same time. A local excitation can provide a researcher access to $\chi(r,t)$ for all values of r .
- Additional efficiency is achieved by enabling the excitation/measurement to be any Hermitian operator, potentially obviating the need to run many circuits.
- Being able to make tailored excitations means that researchers can pinpoint the regime they are interested in and study precisely the excitations of interest, within a single circuit. This is an improvement over rather than having to extract the signal of interest out of the full system response (plus the noise), or having to rely on cancellation between the (noisy) results obtained from several circuit runs.
- Fault tolerant quantum computing comes at a significant qubit overhead cost. An algorithm that has a high tolerance for error and requires 1 fewer qubit will enable earlier hardware calculation of response functions.

We have added these points to the discussion section.

As a demonstration that this method is applicable to any model, we've applied it to obtaining the Green's function of the 6-site 1D Hubbard model (requiring 12 qubits) which is not trivially integrable; that is, no mapping onto a free fermionic problem exists. Unfortunately, the current hardware devices we have access to cannot yet handle the circuit depth required to perform the calculations.

More in general, I believe that the analysis presented by the authors is potentially interesting, but essentially incomplete. Some specific points are listed below:

We appreciate the reviewer's feedback and the opportunity to improve the manuscript. We have made significant changes based on the feedback, and believe the manuscript is now more complete.

- An analysis of the digital unitary decomposition (Trotterization) error is missing. This is an important point to consider if one must implement general time-dependent evolution terms. Overall, it would be important to see concrete scaling arguments concerning the complexity of

quantum circuits (and quantum measurements) necessary to obtain faithful results as a function of the desired frequency and momentum resolution and/or support, and of course as a function of system size.

Our work centers on the linear response framework outlined in the “Bosonic correlation functions” and “Fermionic correlation functions” sections, where we discuss how to obtain these quantities. While they absolutely rely on time evolution, the type of time evolution used (Trotterization, Quantum Signal Processing, analog evolution) does not come into the analysis. Time evolution and digital unitary decomposition is an active area of research, and we point the reviewer to the reference work by Dalzell (cited in the new version of the manuscript in several places) and references therein for an analysis of several current algorithms. The frequency resolution is determined, in first order, by standard Fourier analysis; the resolution is proportional to the inverse maximum simulation time.

Similarly, the scaling with system size follows that of the method used to implement the time evolution. The one additional issue that arises here is the momentum resolution, which is simply given by the inverse system size.

We do agree that the point regarding measurements should be clarified, and we have done so by providing (in the supplement Sec VI) an analysis of the number of shots needed for a given strength of linear response by the quantum system being simulated.

- For the benefit of the readers, a more explicit, step-by-step derivation of the quantum circuits used in the specific experiments, and their compilation on quantum hardware, could also be helpful.

The derivation of the quantum circuits used in the experiments is rather involved, and was discussed at length in References 47 and 48 (formerly 43 and 44), which also have software for constructing the circuits.

We appreciate the reviewer pointing out that this was not clear, and we have amended the text surrounding the first discussion of the circuits to address the issue.

- The two alternative approaches proposed for fermionic correlations are potentially interesting.

However, both methods seem applicable only to certain restricted classes of Hamiltonians: it would be nice to extend the discussion on this point, and for example clearly identify classes of hard problems that the methods could tackle.

To clarify, the classes of Hamiltonians that those methods can work are not so restricted, and include difficult problems of interest to the community. For post-selection, the Hamiltonians we need to conserve the total particle count. This class of Hamiltonians contains virtually all

interacting molecules and condensed matter Hamiltonians such as the Hubbard model. For the auxiliary operator method, the Hamiltonian only needs to conserve parity (even-oddness) of the particle number, which includes the superconducting case as well. It appears that we did not express this clearly in the text, so we added a paragraph discussing this for the auxiliary operator method. We also computed the 1-particle retarded Green's function for the Hubbard model in a new section to show that our method works for hard problems as well.

Moreover, for the auxiliary operator method the authors propose the use of a parity operator, which has a very non-local support: these observables are usually more challenging to measure in the presence of noise – see e.g., *Nature Physics* 19, 752 (2023) –, as also mentioned by the authors on Page 6. Can this become a limitation to the scalability and noise resilience of the proposed protocol?

We thank the reviewer for this very nice question, and the answer is not simply a yes or no because it depends on the situation. In particular, for the case of the auxiliary operator method, since we are interested in measuring a fermionic operator, the product of the parity and Jordan-Wigner string attached to the operator results in a 1-body Pauli to measure, thus avoiding the difficulty with non-local operators. In this sense, the need to involve the parity operator is actually beneficial.

Generically, one can transform a non-local observable into a local one via a chain of CNOT gates. This would be a tradeoff between the measurement noise and CNOT gate noise of course, but it would only require $O(n)$ number of CNOT gates for an n -local observable.

We briefly mention this in the revised manuscript at the end of page 6.

- Concerning the post-selection method, I find the notation in Eqs. 12 not very clear. Probably, moving some additional material from the SI to the main text could be helpful. Are some of these quantities non-normalized wavefunction components?

The reviewer is right. $\psi = \sum_k \phi_k$ where ϕ_k only consists of k -particle states, and ϕ_k are not normalized. As suggested, we added a more detailed explanation in the manuscript right before Eqs. 14 a-c.

How, and (most importantly) how efficiently, can these be reconstructed from measurements taken on a quantum computer? In general, how do the sampling costs of the post-selection scale with the system size?

We thank the reviewer for their excellent question. We have amended both the manuscript and the supplementary material (Sec VI) in response to this question, providing detailed derivations

of how the resource estimate scales with the system size and the precision. We have found that the sampling cost for the post-selection method scales inverse quartically with the precision, and scales polynomially with the system size i.e. $N_{\text{shot}} = \text{Poly}(n)/\epsilon^4$, which shows that our method is scalable to large system sizes.

- On page 4, the statement “This can be tested by repeating the simulation [...] and checking that the response scales similarly” sounds very heuristic. Can anything more systematic be said? For larger problem instances, it could become crucial to be able to specify the experimental details beforehand.

We appreciate the suggestion, and have performed detailed analysis for the instantaneous driving field, where we predict an upper bound that ensures the non-linear effects do not play a role. The bound is based on the spectral and 1-norms of A and B and the desired resolution. We also add that the spectral norms and 1-norms of A and B scale polynomially with the system size for many applications such as the ones we took as an example in the paper: 1 particle Green’s functions, density-density correlators, current-current correlators both in position and momentum basis. The complete derivation is in the SI (Sec VI), with the final result now stated in the manuscript.

- On page 5, “Since $\mu=5$, for this model the single particle energies are manifestly positive”: how is the procedure generalized when such knowledge is not available?

The purpose of this calculation was to demonstrate the noise resilience of the approach by running a calculation on 8 hardware qubits. In order to make this tenable, the circuit depth had to be minimized – here, this was partially accomplished by removing the state preparation step and starting from the empty state. However, this was done entirely for the hardware demonstration; any particular value of μ for a physical system is guided by physics, and no adjustments need to be made for the linear response method. It arises by choosing the chemical potential to be lower than the lowest energy of the first excited state of the system.

We appreciate that this point was not entirely clear, and we have amended the text to specify why this choice was made.

- On page 6, “To minimize the weight of the measured Pauli string (and thus reduce measurement noise) we perform the measurement on the 1st qubit”. Could this be a limitation in general (e.g., reveal boundary effects)?

Similar to the above, this choice was made to improve the calculation for near-term hardware; it is not a limitation of the method. In the example, we excited the system with a momentum mode. Since the model is translationally invariant, the output state will also have the same momentum. In Supplementary Material IV.A, we show that measuring just one qubit is enough to obtain the momentum space Green’s function. Measuring on the first qubit simply removes the Pauli Z string.

More generally, any operator of choice can be measured.

- Page 6, is there any practical explanation for the results to be significantly noisier for $\delta=0.4$? Are for instance the quantum circuits more complex?

This is a good question. We did not find any fundamental explanation for the results for $\delta=0.4$ to be more noisy than the others. The circuit structure is identical to the other values of δ . Rather, we expect that this is due to the strong variation in the day-to-day performance of the IBM quantum hardware. We empirically observe better performance some days than others, even when running the exact same quantum circuits.

- In all experimental results on the SSH model, I think a clear comparison (e.g., in the plots) with the expected, exact results is missing.

In Fig. 2, the exact results are shown as red lines on the false-color plots. Following the reviewer's suggestion, we have also added the lines indicating the exact results in the remaining experimental result figures.

- The discussion at the end of the section "Green's function of the SSH model", concerning Fourier analysis and the impact of noise, is potentially interesting and could become a strong point in favor of the proposed method. Can it be expanded and supported with some more formal studies?

We thank the reviewer for this suggestion. We have formalized the discussion of the impact of noise on the Fourier transformed spectra. In short, we find that any amount of noise that has real space structure (e.g. due to different circuit structures for each real space point) leads to leakage between the various momentum channels that is proportional to the noise structure at the difference between the momentum channels:

$$G_k^{\text{measured}}(\omega) = \frac{1}{\sqrt{n}} \sum_p f_{k-p} G_p^{\text{exact}}(\omega)$$

where f_{k-p} is the $k-p$ component of the Fourier transformed real-space noise structure that arises from the noise being different for different position-space runs.

Simple shot noise, or any noise that is not structured in momentum space does not lead to the same leakage. Rather, the un-structured noise simply adds, washing out the features of the physical system.

We provide a detailed derivation in the SI, and have reproduced the final result in the main text.

- I do not really understand the focus (starting from the title) on the “quantum-inspired” aspect of the method, namely the attempt to propose this also as an alternative solution for classical simulations. One could have the impression that the authors themselves have doubts on the scalability and applicability of the method on quantum computers. The idea of simulating time evolution on classical computers, as opposed to, e.g., approaches based on the Lehmann formula, is in fact quite natural: in practice, however, time evolution is usually hard to realize on classical computers. The authors should support their statements with (1) a more extensive survey in the classical state-of-the-art, identifying the challenges that their method could solve (2) some more concrete computational cost and scaling arguments in the classical domain (corroborating very generic statements like “it is likely to be much more efficient than currently used methods.”

The intent of the “quantum-inspired” aspect of the method was to point out that obtaining bosonic and fermionic response functions via functional differentiation is not limited to its use on quantum computers. In particular, while the bosonic correlation functions are already calculated on classical computers in this manner, no equivalent method exists for fermionic correlation functions (to the best of our knowledge).

Moreover, the classical calculation of correlation functions can necessitate the explicit knowledge of all the eigenstates of the system, requiring full diagonalization of the many-body Hamiltonian. This need arises quite often, and leads e.g. to the computational expense for the GW formulation for electronic structure. Time evolution can be a competitive alternative to full diagonalization for obtaining response functions, as many reasonable time evolution classical algorithms exist for various many-body methods including DMRG, exact diagonalization, DMFT, and beyond. Indeed, there is active development in these areas, precisely in order to avoid the Lehmann approach. We have added a brief discussion of this development in the classical domain to the manuscript in the introduction, and have stricken the more generic statements, including amending the title.

The most common way to compute response functions classically is to do so directly in the frequency domain via Green’s function-based approaches. But for two-particle and higher responses, one needs to solve complicated Beth-Salpeter-like equations, with the reducible or irreducible vertex functions (which often are not easily found). The functional derivative approach avoids this step, by naturally including the vertex corrections, and this can be a significant advantage, even for numerically exact methods such as quantum Monte Carlo methods.

- Overall, I believe the quality of the exposition could be largely improved, particularly to make the whole discussion more accessible to a broader audience. A few examples:

- On Page 2, for instance, a statement like “using the convolution theorem” could be accompanied with a reference, or the definition of X_t (part of B) seems to implicitly assume a Jordan-Wigner mapping, which might confuse non-experts.

The reviewer is right, we amended the manuscript with the definition of Jordan-Wigner mapping. The convolution theorem can be found in any textbook covering Fourier analysis.

- Competing approaches (e.g., the Hadamard test) are never really defined along the discussion.

We thank the reviewer for providing us with the opportunity to improve our comparisons made in the manuscript, which was also requested by the other reviewers. In response, we have now defined the Hadamard test and made our comparisons more clear. The revised manuscript has a restructured introduction, and a new section outlining the Hadamard test.

- The final paragraph of the “Bosonic (commutator) correlation functions” is very hard to follow. What are the “other approaches” or the embedding techniques the authors refer to?

We have amended this text in order to make it clear what we are referring to.

- “Superconducting terms” (page 4) might not tell much to non-specialists.

We have adjusted the text so that it is clear that we refer to pair-creation and pair-annihilation terms.

- Page 6, typo “ain a single Trotter step”.

Thanks to the reviewer for spotting this typo. We corrected it.

- Page 6, “ibm_auckland”: the authors could say something more about the processor (technology, provider, number or qubits, topology, etc...) -- not every reader might recognize the name.

We have added a description of the hardware when it is mentioned, in addition to a hardware connectivity diagram in the supplemental information.

- Page 6, “form of the quantum circuit shown in panel b”: I think the authors mean Fig 2, but a precise reference is missing.

We have corrected the text.

- Page 7, the sentence “This is enabled on the quantum circuit level...as opposed to the Hadamard test which applies B on the state” is a bit obscure.

With the sentence we mean that the way we apply the operator B is different compared to the Hadamard test. We calculate a correlation function of form $\langle A(t)B \rangle$ by applying $\exp(-i \eta B)$ and it allows us to apply linear combinations of Pauli strings in one go, whereas the Hadamard test calculates it by applying B directly and it limits it to applying only unitary operators each time. We have clarified this section.

Reviewer #3 (Remarks to the Author):

In their manuscript, Kökcü et al. introduce a method to retrieve linear response functions from a quantum simulator. Unlike other existing methods that rely on the computation of correlation functions, here the authors propose to implement in the quantum simulation of a many-body system an actual perturbation to the dynamics and then measure the response of the system in the signal of an operator of interest. The correlation function is then retrieved from the measured signal by taking a functional derivative w.r.t. the performed perturbation. According to the authors, this represents a shift in paradigm that brings a number of benefits w.r.t. to existing methods. For example, it allows for frequency and momentum selectivity, which can lead to a reduction in circuit noise for a number of relevant cases, additionally, the method is ancilla free and it can be used to compute correlation functions of non-unitary operators. To illustrate their method, the authors compute Green's functions of a charge density wave based on the Su-Schrieffer-Heger model using the `ibm_auckland` quantum computer.

In general, the manuscript is well-written and clear, and the derivations are sound and support the performed analysis.

We thank the reviewer for their supportive comments.

I am, however, unsure of the significance of the result and fail to see the shift of paradigm that the authors claim. To justify the relevance of the introduced framework the authors state a number of ambiguous statements that, in my view, fail to convincingly show the disruptive character of their method. For example, the authors say that

“The measurement process is truly a part of the simulation as an experimental driving field. This is in contrast to [...] other competing approaches, where the simulation is limited to the system, and the desired observables are extracted either outside of the system qubits and/or from a large excitation.”

It is unclear to me what is meant here by “the measurement process is part of the system” or by “experimental driving field”. Similarly, what is it meant by “observables are extracted outside of the system”? Or why is this worse than the approach here?

The authors also say that “this shift in perspective and methodology enables a much broader set of observables to be envisioned and easily calculated, and enables a direct connection to experiment” What is a direct connection to experiment?

Why are these set of observables easier to calculate? These claims need to be clarified and supported with concrete evidence to establish the significance of the proposed method.

Additionally, the authors state that “it relies almost entirely on time evolution, a task for which quantum computers are naturally suited.” Again, this sounds very ambiguous and fails to show with precision what advantage exactly this brings. Are other approaches not based on time evolution? Are they based on some task that quantum computers are not good at?

The fact that the protocol does not make use of an ancillary system is also stated as an advantage of the method, as it does not require maintaining coherence between the ancilla and the system. This, however, does not seem a significant difference in practice as the current method still requires maintaining coherence between the N qubits of the system, which is just one less qubit than ancilla-based methods that require maintaining coherence between $N+1$ qubits.

The list of hand-waving statements that attempt to convince the reader of a radical superiority of the introduced method w.r.t. to existing methods goes on, but I think my point is clear. The manuscript lacks a comprehensive comparative analysis with existing methods for retrieving linear response functions. The authors make bold claims about the advantages of their approach but do not provide a clear comparison with other approaches based on different principles. The advantages of the method over existing techniques need to be better demonstrated and supported with specific examples to validate the claims.

We appreciate the note that the language was unclear. In framing the discussion this way, we had intended to make a distinction between two cases:

- Simulating the quantum system, and with additional quantum circuit structure, obtain the desired quantity. The Hadamard test approach is a representative of this approach – the system qubits are evolved under their proper Hamiltonian, but ancillary qubits are used to make the desired excitations and collect the response function.
- Simulating the experiment. Response functions are obtained by exciting the system with a driving field, and measuring the change in some observable. No ancillary system is used.

The work we present is the latter type, which has not yet been presented in the quantum computing literature.

As for why approach is superior, we reproduce the response to reviewer B here:

When we get to 10s or 100s of qubits, and we manage to prepare an interesting state of some form, we need to be able to measure its properties. Generically, time evolution is easier than preparing the quantum state – it is in the BQP complexity class, in contrast to state preparation which is in QMA. The method proposed here will enable researchers to efficiently obtain a number of dynamical properties. The fact that it is efficient is important because we expect quantum computation to remain costly for quite some time – this is true today on NISQ hardware, and will also be true for early fault tolerant quantum computing. Thus, if the properties can be obtained in as few circuits as possible, and with a higher tolerance for error, this is beneficial. Our proposed algorithm achieves precisely this.

To be more specific,

- *Efficiency is achieved by enabling the measurement of many correlation functions at the same time. A local excitation can provide a researcher access to $\chi(r,t)$ for all values of r .*
- *Additional efficiency is achieved by enabling the excitation/measurement to be any Hermitian operator, potentially obviating the need to run many circuits.*
- *Being able to make tailored excitations means that researchers can pinpoint the regime they are interested in and study precisely the excitations of interest, within a single circuit. This is an improvement over rather than having to extract the signal of interest out of the full system response (plus the noise), or having to rely on cancellation between the (noisy) results obtained from several circuit runs.*
- *Fault tolerant quantum computing comes at a significant qubit overhead cost. An algorithm that has a high tolerance for error and requires 1 fewer qubit will enable earlier hardware calculation of response functions.*

In order to make these points, we have rewritten the discussion section.

Finally, we appreciate the reviewer comment that a more clear comparison is needed, and we have added this to the revised manuscript. In particular, *we have added a section explaining the capabilities of the Hadamard test, and also added an analysis of the error obtained from the Hadamard test in Fig 3C. There are other existing methods of course, but we have not compared with these because we do not have enough common ground with them; they are methods that either are suited for fault tolerant quantum computers, or rely on variational principles (which cannot be simply analyzed).*

Other minor comments:

In the time-dependent expression for $A(t)$ after Eq. (3), a minus sign is missing in the exponential

Thanks to the reviewer for spotting this typo. We have fixed it.

In the definition of B before Eq. (5) a factor of 2 is missing in front of the cosine after the second equality

Thanks to the reviewer for spotting this typo. We have fixed it.

In conclusion, I find the manuscript's proposed method for retrieving linear response functions in quantum simulators to be interesting and well-executed.

Thanks to the reviewer for their supportive comments.

However, the significance of the results and the claimed shift in paradigm remain unclear due to the lack of concrete evidence and comparative analysis with existing methods.

In the revised manuscript, we have added a clear comparison to the competing methods. We also amended the text by adding a complexity analysis of our method for position, momentum and frequency definite cases.

While this is probably a valuable addition to the toolbox of quantum simulation techniques for response theory, I believe, it lacks the significance required for publication in Nat. Commun. I, thus, recommend addressing the concerns raised above and submitting the work to a more specialized journal where the potential value of this method can be better appreciated within a more focused scientific community.

We respectfully disagree with the reviewer regarding the significance of this work. This is because

- We have presented a different way of thinking about quantum computing simulation of systems in physics and chemistry by bringing in perspectives from these fields, and in particular from the experimental sectors. This new approach is both more efficient and more accurate than previous methods, a significant advance.
- In order to eventually make quantum computing useful for solving problems in physics and chemistry, being able to connect to experiments is critical. We propose that doing a quantum simulation *of the experiment* is precisely the way to do this. Not only is it the most direct way to do so, it also seems to be the most efficient and accurate way to extract important information from the quantum system.
- In doing so, we are illustrating how quantum computation is more relevant to the larger physics and chemistry communities interested in solving the dynamics of the many-body problem, and calling attention to the nascent capabilities of quantum simulation.
- In addition to bridging the divide between quantum simulation and experiment, we have shown that fermionic correlation functions can be directly captured with this method as well, which was previously unknown in both quantum and classical simulation (to the best of our knowledge). Alternative methods either integrate out the fermions and work with auxiliary variables (quantum Monte Carlo methods), or evolve the wavefunction/density matrix in time and compute explicit expectation values (matrix-product methods and exact diagonalization). We are providing a third way to solve these problems, which is an important advance.

- Moreover, we have demonstrated that with this approach, a number of advantages can be gained that lie purely in the quantum computing realm, including decreased noise sensitivity, decreased ancilla count, and potential specificity in the circuits to run.
- As concrete evidence of these capabilities, we were able to obtain spectra that previously have not been demonstrated on hardware.

With these, we have made a number of significant improvements, in several directions, over the current state of the art in this important problem that cuts across the boundaries of all scientific fields involved in computing dynamical properties of quantum systems. We have also made amendments to the manuscript in order to make these points more clear.

REVIEWERS' COMMENTS

Reviewer #1 (Remarks to the Author):

The authors addressed my comments very satisfactorily. I recommend publication of this work as is.

Reviewer #2 (Remarks to the Author):

I thank the authors for addressing my comments and suggestions. In this revised manuscript, they made substantial improvements in response to almost all the points raised. I would therefore recommend publishing the manuscript in its present form, or with only minor amendments.

However, although the results are technically sound, I am not yet fully convinced that the proposed protocol really represents a highly significant and broadly applicable “paradigm shift” for near term quantum computers. In fact, the experimental results are comparable (or smaller) in scale and complexity to current state-of-the-art.

At the algorithmic level, the method proposed by the authors is certainly a nice addition to the digital quantum simulations toolbox and presents convincingly nice features. However, it is not fully clear whether its overall costs leave enough room to critically enhance the size of feasible simulations: for instance, the relatively high sampling requirements (which are inversely proportional to both the square of the nonlinear and the shot noise error contributions, and directly proportional to a polynomial in the system size) might quickly become impractical. In fact, even though everything remains formally efficient in the asymptotic regime, there have been multiple cases in which even for modest-degree polynomials the actual overhead is substantial (see e.g., Phys. Rev. A 92, 042303 (2005) for chemistry applications).

I think the work could meet the criteria for Nature Communications if the authors presented a larger scale implementation, or if a thorough analysis revealed that this would be feasible (in terms of actual number of circuits, shots, and noise levels) with current quantum devices. Otherwise, I would tend to agree with Referee #3, and recommend publication in a more specialized journal.

Reviewer #3 (Remarks to the Author):

Having thoroughly reviewed the authors' responses to both my inquiries and those of the other referees, along with their revised manuscript, it is evident that the authors have undertaken an important effort to contextualize their work in relation to alternative approaches for measuring response functions. Their analysis extends to scrutinizing errors and efficiency, with a clear articulation of the advantages of their approach with respect to other existing methods, as specifically requested by myself and another referee. I am confident that the extensive modifications made by the authors have significantly enhanced the manuscript's quality, providing a clearer understanding of the significance of their results. Consequently, I am persuaded by the overall response of the authors and am inclined to recommend the publication of this work in its current form.

Reviewer #1 (Remarks to the Author):

The authors addressed my comments very satisfactorily. I recommend publication of this work as is.

We thank the reviewer for their comments. Their input has helped to significantly improve the manuscript.

Reviewer #2 (Remarks to the Author):

I thank the authors for addressing my comments and suggestions. In this revised manuscript, they made substantial improvements in response to almost all the points raised. I would therefore recommend publishing the manuscript in its present form, or with only minor amendments.

We thank the reviewer for their positive comments.

However, although the results are technically sound, I am not yet fully convinced that the proposed protocol really represents a highly significant and broadly applicable “paradigm shift” for near term quantum computers. In fact, the experimental results are comparable (or smaller) in scale and complexity to current state-of-the-art.

The referee is correct that the experimental results are comparable to the state of the art. As we outline in the manuscript (and provide further details including resource estimates below), the advantage of our proposed protocol is over competing techniques, and these advantages either remain or get stronger as the system size increases.

In our last revision, following the reviewer’s suggestion, we have provided resource estimates and how these scale with the system size.

At the algorithmic level, the method proposed by the authors is certainly a nice addition to the digital quantum simulations toolbox and presents convincingly nice features. However, it is not fully clear whether its overall costs leave enough room to critically enhance the size of feasible simulations: for instance, the relatively high sampling requirements (which are inversely proportional to both the square of the nonlinear and the shot noise error contributions, and directly proportional to a polynomial in the system size) might quickly become impractical. In fact, even though everything remains formally efficient in the asymptotic regime, there have been multiple cases in which even for modest-degree polynomials the actual overhead is substantial (see e.g., Phys. Rev. A 92, 042303 (2005) for chemistry applications). I think the work could meet the criteria for Nature Communications if the authors presented a larger scale implementation, or if a thorough analysis revealed that this would be feasible (in terms of actual number of circuits, shots, and noise levels) with current quantum devices. Otherwise, I would tend to agree with Referee #3, and recommend publication in a more specialized journal.

We thank the reviewer for their reasonable request for a resource estimation. Unfortunately, running calculations on a larger quantum hardware system is not feasible, so we follow their alternative suggestion; we calculate feasibility and resource estimates, and perform classical simulations of larger systems.

In what follows, as the referee suggested, we analyze how the number of shots scales with system size n and desired accuracy ϵ_{Total} , and show that an advantage of our proposed protocol remains. Our salient points are:

1. For the primary competing method (the Hadamard test),
 - a. $N_{shots} = \epsilon_{Total}^{-2}$ for a position-definite response function (or similar)
 - b. $N_{shots} = \epsilon_{Total}^{-2} n$ for a momentum-definite response function (or similar)
 - c. The number of individual circuits needed to run for a momentum-definite response function (or similar) scales with n
2. For our linear response protocol,
 - a. $N_{shots} = \epsilon_{Total}^{-2} \eta^{-2}$ for any response function
 - b. Empirically, we observe that the bound on the non-linear error is extremely loose
 - c. The Trotter error bound is also loose, and can be improved (our analysis used the simplest form possible)
3. Combining these, we show
 - a. Linear response shows an n^2 factor of improvement in the total number of shots needed for momentum definite response functions, or others where the operators are not a simple local ones
 - b. As discussed in the manuscript, for situations where the response function is local, many (or order n) are obtained from a single circuit run.
 - c. For empirically-determined bounds on the non-linear error and Trotter error for 16 sites, the linear response protocol requires fewer shots than the Hadamard test in all cases considered.

We detail these considerations below, and in the revised supplement. With this, we believe we have addressed the referees final request.

Analysis for Number of Shots per Circuit

We start with the analysis for the number of shots required per circuit, and we compare our method to the Hadamard method. On a perfectly fault tolerant quantum computer, the Hadamard test only has shot noise. Calculating a position dependent response function to ϵ_{Total} accuracy via Hadamard test requires $N_{shots} = \epsilon_{Total}^{-2}$ shots. In the momentum dependent case, since the response function is going to be a Fourier transformation of n position dependent response functions:

$$G_k(t) = \frac{1}{\sqrt{n}} \sum_r e^{ikr} G_r(t) ,$$

each of these positional response functions must be calculated to $\epsilon_{Total}/\sqrt{n}$ accuracy, which leads to $N_{shots} = \epsilon_{Total}^{-2} n$ for each circuit. We will be referring to these throughout our response.

The error in our linear response method separates into three categories: non-linear error, Trotter error for the driving field, and the measurement shot noise. The system size n dependence does not occur explicitly in any of these: the errors depend on the non-linear error threshold and operator norms $\|A\|$ and $\|B\|$. In the manuscript, we kept the freedom to choose these operators, and assumed that the norms are Poly(n) where n is the system size. However, for many cases in physics these operators are normalized with the system size, and their norms become $O(1)$. With this kept in mind, from Eqs. S34-S46-S52, each of these errors can be given in terms of the signal amplitude η as $\epsilon_{NL} = 2\eta e^{2\eta}$, $\epsilon_{Trot} = \alpha(B)\eta$ and $\epsilon_{meas} = 1/(\eta\sqrt{N_{shots}})$, with

$$\epsilon_{Total} = \epsilon_{NL} + \epsilon_{Trot} + \epsilon_{meas} .$$

η is determined by the desired ϵ_{NL} and ϵ_{Trot} , and the number of shots is determined via keeping

$$\epsilon_{meas} \text{ under a threshold value by setting } N_{shots} = \epsilon_{Total}^{-2} \eta^{-2} .$$

The η values given via these theoretical error bounds is rather small, and leads to a very large number of shots. Below, we discuss that both of these theoretical error bounds are very loose for physics simulations, which is the main scope of our method. The looseness of these bounds lead to the fact that our method performs much better than Hadamard test for momentum definite cases.

Looseness of the Nonlinearity Error Bound ϵ_{NL}

Empirically (and experimentally) speaking, observing non-linear response functions tends to require relatively large driving fields. To illustrate this, here we show a plot of the $\mathbf{k}=\pi$ transverse spin correlation function for the antiferromagnetic Heisenberg model with 16 spins (much larger than was presented in the manuscript):

We calculated the correlation function exactly by using numpy to obtain the non-linearity error ϵ_{NL} for various η values and compare this to the analytic bounds obtained. It can be seen from the plots that, because the correlation function is measured on a specific state of the model rather than the one that yields the operator norm, we have $\epsilon_{NL} \ll 2\eta e^{2\eta}$, and therefore we can choose the η value a lot larger than the bound we provided in the supplementary material (Eqs. S53-S54).

We next calculate the requirements for both our method and the Hadamard test. For this response function, the operators A and B are the same and given as the following:

$$\mathbf{A} = \mathbf{B} = \frac{1}{n} (S_1^Z - S_2^Z + S_3^Z - S_4^Z \pm \dots + (-1)^{n-1} S_n^Z)$$

where $n = 16$ is the system size. The operator norms are $\|\mathbf{A}\| = \|\mathbf{B}\| = 1$, and since they consist of commuting Pauli terms, there is no Trotter error in the application of the signal with B. Thus, the linear-response method has total error $\epsilon_{Total} = \epsilon_{NL} + \epsilon_{meas}$ for this case, and requires only a single circuit run. The Hadamard test, however, requires $n = 16$ circuit runs. Thus for each circuit, it requires $\epsilon_{Total}/\sqrt{n}$ accuracy, which is $\epsilon_{Total}/4$ in this particular case and would require smaller accuracy as the system size grows.

For this specific model, the number of shots required by our method and the Hadamard test method together with the choices of η values with respect to both theoretical and the empirical error bounds are given in the following table. We use the empirical bounds on η .

	Hadamard $N_{shots} = \epsilon_{Total}^{-2} n$	Linear Response $N_{shots} = \epsilon_{Total}^{-2} \eta^{-2}$	η from Theoretical bound	η from empirical bound
$\epsilon_{Total} = 10^{-2}$	$n \times 10^4 = 16 \times 10^4$ shots per circuit $n = 16$ circuit runs $n^2 \times 10^4 = \mathbf{256 \times 10^4}$ shots in total	10^4 shots 1 circuit run 10^4 shots in total	0.005	1
$\epsilon_{Total} = 10^{-3}$	$n \times 10^6 = 16 \times 10^6$ shots per circuit $n = 16$ circuit runs $n^2 \times 10^6 = \mathbf{256 \times 10^6}$ shots in total	5×10^6 shots 1 circuit run $\mathbf{5 \times 10^6}$ shots in total	0.0005	0.2

As one can see, even for a modest system size $n=16$, our method requires many fewer shots in total compared to the Hadamard test for this correlation function, and this advantage grows for larger system sizes: for each circuit, we require a factor of $O(n)$ less shots, and in total we require a factor of $O(n^2)$ less shots.

Looseness of the Trotter Error Bound ϵ_{Trot}

For many physics systems, the response function is normalized such that the operator norms $\|A\|$, $\|B\|$ and the commutator norm $\alpha(B)$ are independent of the system size. One notable exception is the momentum dependent fermion Green function. Due to the normalization of the Green's function, the operator norms $\|A\|$ and $\|B\|$ again become 1, however the commutator norm of the operator B scales with the system size: $\alpha(B) = n$. This makes the theoretical Trotter error bound scale as $O(n \eta)$ where n is the system size. To overcome this, we propose application of $\exp(i \eta B)$ operator in not one but multiple Trotter steps. When r Trotter steps are used, the error scales as $O(\alpha(B)/r)$. This means that for this particular B , if we apply our signal in n trotter steps, none of the three errors depend on the system size, leading to an η value independent of the system size.

However, even without this change, this error bound is also found to be loose for systems of interest. Here we show the Trotter error for the **spinless half filled Hubbard model with 16 sites** with momentum value $k = 2\pi/16$, where we apply our signal with a **single Trotter step**:

As can be seen empirically from the figures above, Trotter error is approximately three orders of magnitude smaller than the theoretical error bound. This reduces the number of shots since $N_{shots} = 1/(\eta^2 \epsilon_{Total}^2)$ and leads to the following estimated shot counts for both methods (using the empirical bounds on η):

	Hadamard $N_{shots} = \epsilon_{Total}^{-2} n$	Linear Response $N_{shots} = \epsilon_{Total}^{-2} \eta^{-2}$	η from Theoretical bound	η from empirical bound
$\epsilon_{Total} = 10^{-2}$	$n \times 10^4 = 16 \times 10^4$ shots per circuit $n = 16$ circuit runs $n^2 \times 10^4 = 2.56 \times 10^6$ shots in total	62500 shots 1 circuit run 62500 shots in total	$\frac{\epsilon_{Total}}{n} = 6.25 \times 10^{-4}$	0.4
$\epsilon_{Total} = 10^{-3}$	$n \times 10^6 = 16 \times 10^6$ shots per circuit $n = 16$ circuit runs $n^2 \times 10^6 = 2.56 \times 10^8$ shots in total	10^8 shots 1 circuit run 10^8 shots in total	$\frac{\epsilon_{Total}}{n} = 6.25 \times 10^{-5}$	0.1

The resource estimate for our method becomes comparable to the Hadamard test only for $\epsilon_{Total} = 10^{-3}$ for this system size ($n = 16$), and the number of shots required for the linear response is still a factor of 2 better. With the incorporation of multiple Trotter steps, this number of shots requirement can be kept under control for larger system sizes, whereas the Hadamard test would require significantly more resources than our method.

Analysis for Number of Circuits to Run

We finish our analysis with the number of circuits required to measure certain response functions. In this part, our method has an additional advantage that *does* scale with system size when considering the number of circuits to run. Often, one does not want to calculate one but many correlation functions $\langle A_i(t) B_j \rangle$. In this case it becomes important whether the operators A_i commute with each other or not. In the case the operators A_i do not commute, such as the fermionic Green's functions, then they cannot be measured at the same time, and the number of circuits to be run is the same as with the Hadamard test. When the operators are bosonic such as spin-spin correlations and the density-density correlations, A_i commute, and therefore, they can all be measured at the same time. This allows us to reduce the number of circuits by a factor of n (system size) compared to Hadamard test. As a concrete example: the density-density correlator we calculated for the SSH model (because the model is translationally invariant) only needs **2** circuits (one for the real part, one for the imaginary part) to calculate all correlation functions. This is independent of the system size. The Hadamard test requires **$2n$** circuits, which would be 200 circuits for a 100 spin system. If the model is not translationally invariant, we need to run **$2n$** circuits while the Hadamard test requires **$2n^2$** circuits. In this case for a 100 spin system our method would require 200 circuits, while the Hadamard test requires 20,000 circuits.

Another case to estimate the resources for is the momentum-space fermionic Green's function. Here, even though the total number of circuits to obtain the Green's function for every individual momentum value is the same as the Hadamard test and proportional to the system size n , there are two important advantages of our method:

1. If one desires to calculate the Green's function of a translationally invariant system for a finite momentum interval with K momentum values, our method requires **$2K$** circuits, while the Hadamard test requires **$2n$** circuits,
2. We have shown that our method is more noise robust than the Hadamard test for NISQ machines. Thus, even when the operators A_i do not commute with each other, our method requires fewer or equivalent resources, and produces better results than the Hadamard test in noisy quantum hardware.

Reviewer #3 (Remarks to the Author):

Having thoroughly reviewed the authors' responses to both my inquiries and those of the other referees, along with their revised manuscript, it is evident that the authors have undertaken an important effort to contextualize their work in relation to alternative approaches for measuring response functions. Their analysis extends to scrutinizing errors and efficiency, with a clear articulation of the advantages of their approach with respect to other existing methods, as specifically requested by myself and another referee. I am confident that the extensive modifications made by the authors have significantly enhanced the manuscript's quality, providing a clearer understanding of the significance of their results. Consequently, I am persuaded by the overall response of the authors and am inclined to recommend the publication of this work in its current form.

We thank the reviewer for their comments. Their input has helped to significantly improve the manuscript.